# Memory Based Trajectory-conditioned Policies for Learning from Sparse Rewards

**Yijie Guo**[1]    **Jongwook Choi**[1]    **Marcin Moczulski**[2]    **Shengyu Feng**[1]
**Samy Bengio**[2]    **Mohammad Norouzi**[2]    **Honglak Lee**[2,1]
[1]University of Michigan    [2]Google Brain
{guoyijie,jwook,shengyuf}@umich.edu   moczulski@google.com
{bengio,mnorouzi,honglak}@google.com

## Abstract

Reinforcement learning with sparse rewards is challenging because an agent can rarely obtain non-zero rewards and hence, gradient-based optimization of parameterized policies can be incremental and slow. Recent work demonstrated that using a memory buffer of previous successful trajectories can result in more effective policies. However, existing methods may overly exploit past successful experiences, which can encourage the agent to adopt sub-optimal and myopic behaviors. In this work, instead of focusing on good experiences with limited diversity, we propose to learn a trajectory-conditioned policy to follow and expand diverse past trajectories from a memory buffer. Our method allows the agent to reach diverse regions in the state space and improve upon the past trajectories to reach new states. We empirically show that our approach significantly outperforms count-based exploration methods (parametric approach) and self-imitation learning (parametric approach with non-parametric memory) on various complex tasks with local optima. In particular, without using expert demonstrations or resetting to arbitrary states, we achieve the state-of-the-art scores under five billion number of frames, on challenging Atari games such as Montezuma's Revenge and Pitfall.

## 1   Introduction

Deep reinforcement learning (DRL) algorithms with parameterized policy and value function have achieved remarkable success in various complex domains [32, 49, 48]. However, tasks that require reasoning over long horizons with sparse rewards remain exceedingly challenging for the parametric approaches. In these tasks, a positive reward could only be received after a long sequence of appropriate actions. The gradient-based updates of parameters are incremental and slow and have a global impact on all parameters, which may cause catastrophic forgetting and performance degradation. Many parametric approaches rely on recent samples and do not explore the state space systematically. They might forget the positive-reward trajectories unless the good trajectories are frequently collected.

Recently, non-parametric memory from past experiences is employed in DRL algorithms to improve policy learning and sample efficiency. Prioritized experience replay [45] proposes to learn from past experiences by prioritizing them based on temporal-difference error. Episodic reinforcement learning [43, 22, 28], self-imitation learning [36, 19], and memory-augmented policy optimization [27] build a memory to store past good experience and thus can rapidly latch onto past successful policies when encountering with states similar to past experiences. However, the exploitation of good experiences within limited directions might hurt performance in some cases. For example on Montezuma's Revenge (Fig. 1), if the agent exploits the past good trajectories around the yellow path, it would receive the small positive rewards quickly but it loses the chance to achieve a higher score in the long term. Therefore, in order to find the optimal path (red), it is better to consider past experiences in diverse directions, instead of focusing only on the good trajectories which lead to myopic behaviors. Inspired by recent work on memory-augmented generative models [21, 9], we note that generating a

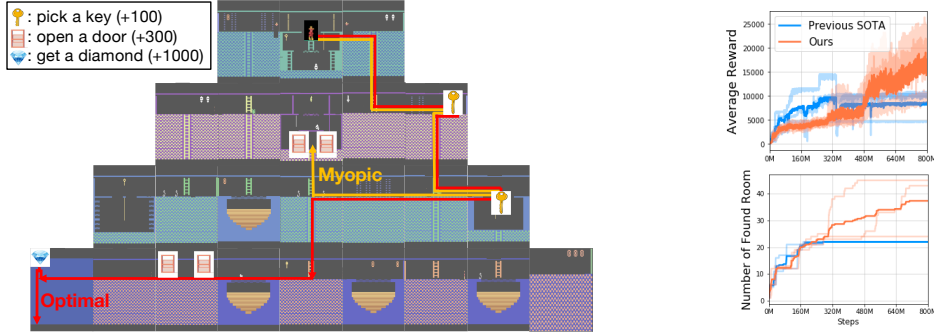

Figure 1: **Left:** The map of the first level in Montezuma's Revenge. We simplify the agent's paths and enlarge some objects to illustrate typical *exploration challenges*. The agent also needs to tackle *control challenges* (e.g., jumping between platforms, avoiding collision with moving enemies and electric fields, etc.), but they are not highlighted here. After getting two keys, the agent can easily expense the keys to open doors in the middle via the yellow path and achieve small incremental rewards, but as each key can only be used once, the agent is unlikely to open doors at the bottom floor to clear the level. The previous SOTA fails to open the last two doors. Ours visits the left-most room at the bottom floor, gets many diamonds, and goes to the next level. **Right:** Comparison to CoEX [13] (previous SOTA) with high-level state embedding. In a challenging setting with random initial delay, without using expert demonstrations or resetting to arbitrary state, ours explores more rooms and achieves a significantly higher score.

new sequence by editing prototypes in external memory is easier than generating one from scratch. In an RL setting, we aim to generate new trajectories visiting novel states by editing or augmenting the trajectories stored in the memory from past experiences. We propose a novel trajectory-conditioned policy where a full sequence of states is given as the condition. Then a sequence-to-sequence model with an attention mechanism learns to 'translate' the demonstration trajectory to a sequence of actions and generate a new trajectory in the environment with stochasticity. The single policy could take diverse trajectories as the condition, imitate the demonstrations to reach diverse regions in the state space, and allow for flexibility in the action choices to discover novel states.

Our main contributions are summarized as follows. (1) We propose a novel architecture for a trajectory-conditioned policy that can flexibly imitate diverse demonstration trajectories. (2) We show the importance of exploiting diverse past experiences in the memory to indirectly drive exploration, by comparing with existing approaches on various sparse-reward RL tasks with stochasticity in the environments. (3) We achieve a performance superior to the state-of-the-art under 5 billion number of frames, on hard-exploration Atari games of Montezuma's Revenge and Pitfall, without using expert demonstrations or resetting to arbitrary states. We also demonstrate the effectiveness of our method on other benchmarks.

## 2 Method

### 2.1 Background and Notation for DTSIL

In the standard RL setting, at each time step $t$, an agent observes a state $s_t$, selects an action $a_t \in \mathcal{A}$, and receives a reward $r_t$ when transitioning to a next state $s_{t+1} \in \mathcal{S}$, where $\mathcal{S}$ and $\mathcal{A}$ is a set of states and actions respectively. The goal is to find a policy $\pi_\theta(a|s)$ parameterized by $\theta$ that maximizes the expected return $\mathbb{E}_{\pi_\theta}[\sum_{t=0}^{T} \gamma^t r_t]$, where $\gamma \in (0, 1]$ is a discount factor. In our work, instead of directly maximizing expected return, we propose a novel way to find best demonstrations $g^*$ with (near-)optimal return and train the policy $\pi_\theta(\cdot|g)$ to imitate any trajectory $g$ in the buffer, including $g^*$. We assume a state $s_t$ includes the observation $o_t$ (e.g., raw pixel image) and a high-level abstract state embedding $e_t$ (e.g., the agent's location in the abstract space). The embedding $e_t$ may be available as a part of $s_t$ (e.g., the physical features in the robotics domain) or may be learnable from $o_{\leq t}$ (e.g., [13, 54] could localize the agent in Atari games, as discussed in Sec. 5). A *trajectory-conditioned policy* $\pi_\theta(a_t|e_{\leq t}, o_t, g)$ (which can be viewed as a goal-conditioned policy and denoted as $\pi_\theta(\cdot|g)$) takes a sequence of state embeddings $g = \{e_1^g, e_2^g, \cdots, e_{|g|}^g\}$ as input for a demonstration, where $|g|$ is the length of the trajectory $g$. A sequence of the agent's past state embeddings $e_{\leq t} = \{e_1, e_2, \cdots, e_t\}$ is provided to determine which part of the demonstration has been followed by the agent. Together with the current observation $o_t$, it helps to determine the correct action $a_t$ to imitate the demonstration. Our goal is to find a set of optimal state embedding sequence(s) $g^*$ and the policy $\pi_\theta^*(\cdot|g)$ to maximize the return: $g^*, \theta^* \triangleq \arg\max_{g,\theta} \mathbb{E}_{\pi_\theta(\cdot|g)}[\sum_{t=0}^{T} \gamma^t r_t]$. We approximately solve this joint optimization

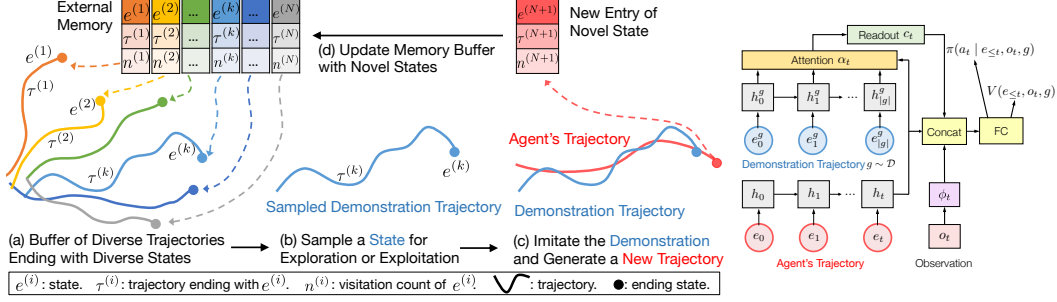

Figure 2: **Left**: Overview of DTSIL. (a) We maintain a trajectory buffer. (b) For each episode, we sample a state from the buffer, (c) imitate the demonstration leading to the sampled state, obtain a new trajectory, (d) update the memory with the new trajectory and gradually expand the buffer. We repeat this process until training goes to the end. **Right**: Architecture of the trajectory-conditioned policy (see details in Sec. 2.3).

problem via the sampling-based search for $g^*$ over the space of $g$ realizable by the (trajectory-conditioned) policy $\pi_\theta$ and gradient-based local search for $\theta^*$. For robustness, we may want to find multiple trajectories with high returns and a trajectory-conditioned policy executing them. We name our method as *Diverse Trajectory-conditioned Self-Imitation Learning* (DTSIL).

## 2.2 Overview of DTSIL

**Organizing Trajectory Buffer**   As shown in Fig. 2(a), we maintain a *trajectory buffer* $\mathcal{D} = \{(e^{(1)}, \tau^{(1)}, n^{(1)}), (e^{(2)}, \tau^{(2)}, n^{(2)}), \cdots\}$ of diverse past trajectories. $\tau^{(i)}$ is the best trajectory ending with a state with embedding $e^{(i)}$. $n^{(i)}$ is the number of times the cluster represented by the embedding $e^{(i)}$ has been visited during training. To maintain a compact buffer, similar state embeddings within the tolerance threshold $\delta$ are clustered together, and an existing entry is replaced if an improved trajectory $\tau^{(i)}$ ending with a near-identical state is found. In the buffer, we keep a single representative state embedding for each cluster. If a state embedding $e_t$ observed in the current episode is close to a representative state embedding $e^{(k)}$, we increase visitation count $n^{(k)}$ of the $k$-th cluster. If the sub-trajectory $\tau_{\leq t}$ of the current episode up to step $t$ is better than $\tau^{(k)}$, $e^{(k)}$ is replaced by $e_t$. Pseudocode for organizing clusters is in the appendix.

**Sampling States for Exploitation or Exploration**   In RL algorithms, the agent needs to *exploit* what it already knows to maximize reward and *explore* new behaviors to find a potentially better policy. For exploitation, we aim at reaching the states with the highest total rewards, which probably means a good behavior of receiving high total rewards. For exploration, we would like to look around the rarely visited states, which helps discover novel states with higher total rewards. With probability $1 - p$, in exploitation mode, we sample the states in the buffer with the highest cumulative rewards. With probability $p$, in exploration mode, we sample each state $e^{(i)}$ with the probability proportional to $1/\sqrt{n^{(i)}}$, as inspired by count-based exploration [50, 7] and rank-based prioritization [45, 16]. To balance between exploration and exploitation, we decrease the hyper-parameter $p$ of taking the exploration mode. The pseudo-code algorithm of sampling the states is in the appendix.

**Imitating Trajectory to State of Interest**   In stochastic environments, in order to reach diverse states $e^{(i)}$ we sampled, the agent would need to learn a goal-conditioned policy [1, 34, 44, 40]. But it is difficult to learn the goal-conditioned policy only with the final goal state because the goal state might be far from the agent's initial states and the agent might have few experiences reaching it. Therefore, we provide the agent with the *full* trajectory leading to the goal state. So the agent benefits from richer intermediate information and denser rewards. We call this *trajectory-conditioned policy* $\pi_\theta(\cdot|g)$ where $g = \{e_1^g, e_2^g, \cdots, e_{|g|}^g\}$, and introduce how to train the policy in detail in Sec. 2.3.

**Updating Buffer with New Trajectory**   With trajectory-conditioned policy, the agent takes actions to imitate the sampled demonstration trajectory. As shown in Fig. 2(c), because there could be stochasticity in the environment and our method does not require the agent to exactly follow the demonstration step by step, the agent's new trajectory could be different from the demonstration and thus visit novel states. In a new trajectory $\mathcal{E} = \{(o_0, e_0, a_0, r_0), \cdots, (o_T, e_T, a_T, r_T)\}$, if $e_t$ is nearly identical to a state embedding $e^{(k)}$ in the buffer and the partial episode $\tau_{\leq t}$ is better than (i.e. higher return or shorter trajectory) the stored trajectory $\tau^{(k)}$, we replace the existing entry $(e^{(k)}, \tau^{(k)}, n^{(k)})$ by $(e_t, \tau_{\leq t}, n^{(k)} + 1)$. If $e_t$ is not sufficiently similar to any state embedding in the buffer, a new entry $(e_t, \tau_{\leq t}, 1)$ is pushed into the buffer, as shown in Fig. 2(d). Therefore we gradually increase the diversity of trajectories in the buffer. The detailed algorithm is described in the supplementary material.

## 2.3 Learning Trajectory-Conditioned Policy

**Policy Architecture**    For imitation learning with diverse demonstrations, we design a trajectory-conditioned policy $\pi_\theta(a_t|e_{\leq t}, o_t, g)$ that should flexibly imitate any given trajectory $g$. Inspired by neural machine translation methods [51, 6], one can view the demonstration as the source sequence and view the incomplete trajectory of the agent's state representations as the target sequence. We apply a recurrent neural network (RNN) and an attention mechanism Bahdanau et al. [6] to the sequence data to predict actions that would make the agent follow the demonstration. As illustrated in Fig. 2, RNN computes the hidden features $h_i^g$ for each state embedding $e_i^g$ ($0 \leq i \leq |g|$) in the demonstration and derives the hidden features $h_t$ for the agent's state representation $e_t$. Then the attention weight $\alpha_t$ is computed by comparing the current agent's hidden features $h_t$ with the demonstration's hidden features $h_i^g$ ($0 \leq i \leq |g|$). The attention readout $c_t$ is computed as an attention-weighted summation of the demonstration's hidden features to capture the relevant information in the demonstration and to predict the action $a_t$. Training is performed by combining RL and supervised objectives as follows.

**Reinforcement Learning Objective**    Given a demonstration trajectory $g = \{e_0^g, e_1^g, \cdots, e_{|g|}^g\}$, we provide rewards for imitating $g$ and train the policy to maximize rewards. For each episode, we record $u$ to denote the index of state in the given demonstration that is lastly visited by the agent. At the beginning of an episode, the index $u$ of the lastly visited state embedding in the demonstration is initialized as $u = -1$, which means no state in the demonstration has been visited. At each step $t$, if the agent's new state $s_{t+1}$ has an embedding $e_{t+1}$ and it is the similar enough to any of the next $\Delta t$ state embeddings starting from the last visited state embedding $e_u^g$ in the demonstration (i.e., $\|e_{t+1} - e_{u'}^g\| < \delta$ where $u < u' \leq u + \Delta t$), then the index of the last visited state embedding in the demonstration is updated as $u \leftarrow u'$ and the agent receives environment reward and positive imitation reward $r_t^{\mathrm{DTSIL}} = f(r_t) + r^{\mathrm{im}}$, where $f(\cdot)$ is a monotonically increasing function (e.g., clipping [32]) and $r^{\mathrm{im}}$ is the imitation reward with a value of 0.1 in our experiments. Otherwise, the reward $r_t^{\mathrm{DTSIL}}$ is 0 (see appendix for an illustration example). This encourages the agent to visit states in the demonstration in a soft-order so that it can edit or augment the demonstration when executing a new trajectory. The demonstration plays a role to guide the agent to the region of interest in the state embedding space. After visiting the last (non-terminal) state in the demonstration, the agent performs random exploration (because $r_t^{\mathrm{DTSIL}} = 0$) around and beyond the last state until the episode terminates, to push the frontier of exploration. With $r_t^{\mathrm{DTSIL}}$, the trajectory-conditioned policy $\pi_\theta$ can be trained with a policy gradient algorithm [52]:

$$\mathcal{L}^{\mathrm{RL}} = \mathbb{E}_{\pi_\theta}[-\log \pi_\theta(a_t|e_{\leq t}, o_t, g)\widehat{A}_t], \tag{1}$$

$$\text{where } \widehat{A}_t = \sum_{d=0}^{n-1} \gamma^d r_{t+d}^{\mathrm{DTSIL}} + \gamma^n V_\theta(e_{\leq t+n}, o_{t+n}, g) - V_\theta(e_{\leq t}, o_t, g)$$

where $\mathbb{E}_{\pi_\theta}$ indicates the empirical average over a finite batch of on-policy samples and $n$ denotes the number of rollout steps taken in each iteration. We use Proximal Policy Optimization [48] as an actor-critic policy gradient algorithm for our experiments.

**Supervised Learning Objective**    To improve trajectory-conditioned imitation learning and to better leverage the past trajectories, we propose a supervised learning objective. We leverage the actions in demonstrations, similarly to behavior cloning, to help RL for imitation of diverse trajectories. We sample a trajectory $\tau = \{(o_0, e_0, a_0, r_0), (o_1, e_1, a_1, r_1) \cdots\} \in \mathcal{D}$, formulate the demonstration $g = \{e_0, e_1, \cdots, e_{|g|}\}$ and assume the agent's incomplete trajectory is the partial trajectory $g_{\leq t} = e_{\leq t} = \{e_0, e_1, \cdots, e_t\}$ for any $1 \leq t \leq |g|$. Then $a_t$ is the 'correct' action at step $t$ for the agent to imitate the demonstration. Our supervised learning objective is to maximize the log probability of taking such actions:

$$\mathcal{L}^{\mathrm{SL}} = -\log \pi_\theta(a_t|e_{\leq t}, o_t, g), \text{ where } g = \{e_0, e_1, \cdots, e_{|g|}\}. \tag{2}$$

## 2.4 Extensions of DTSIL for Improved Robustness and Generalization

DTSIL can be easily extended for more challenging scenarios. Without hand-crafted high-level state embeddings, we can combine DTSIL with state representation learning approaches (Sec. 5.1). In highly stochastic environments, we modify DTSIL to construct and select proper demonstrations (Sec. 5.2). In addition, DTSIL can be extended with hierarchical reinforcement learning for generalization over multiple tasks (Sec. 5.3). See individual sections for more details.

# 3 Related Work

**Imitation Learning**   The goal of imitation learning is to train a policy to mimic a given demonstration. Many previous works achieve good results on hard-exploration Atari games by imitating human demonstrations [23, 41]. Aytar et al. [3] learn embeddings from a variety of demonstration videos and proposes the one-shot imitation learning reward, which inspires the design of rewards in our method. All these successful attempts rely on the availability of human demonstrations. In contrast, our method treats the agent's past trajectories as demonstrations.

**Memory Based RL**   An external memory buffer enables the storage and usage of past experiences to improve RL algorithms. Episodic reinforcement learning methods [43, 22, 28] typically store and update a look-up table to memorize the best episodic experiences and retrieve the episodic memory in the agent's decision-making process. Oh et al. [36] and Gangwani et al. [19] train a parameterized policy to imitate only the high-reward trajectories with the SIL or GAIL objective. Unlike the previous work focusing on high-reward trajectories, we store the past trajectories ending with diverse states in the buffer, because trajectories with low reward in the short term might lead to high reward in the long term. Badia et al. [5] train a range of directed exploratory policies based on episodic memory. Gangwani et al. [19] propose to learn multiple diverse policies in a SIL framework but their exploration can be limited by the number of policies learned simultaneously and the exploration performance of every single policy, as shown in the supplementary material.

**Learning Diverse Policies**   Previous works [20, 17, 42] seek a diversity of policies by maximizing state coverage, the entropy of mixture skill policies, or the entropy of goal state distribution. Zhang et al. [56] learns a variety of policies, each performing novel action sequences, where the novelty is measured by a learned autoencoder. However, these methods focus more on tasks with relatively simple state space and dense rewards while DTSIL shows experimental results performing well on long-horizon, sparse-reward environments with a rich observation space like Atari games.

**Exploration**   Many exploration methods [46, 2, 12, 50] in RL tend to award a bonus to encourage an agent to visit novel states. Recently this idea was scaled up to large state spaces [53, 7, 38, 11, 39, 10]. Intrinsic curiosity uses the prediction error or pseudo count as intrinsic reward signals to incentivize visiting novel states. We propose that instead of directly taking a quantification of novelty as an intrinsic reward, one can encourage exploration by rewarding the agent when it successfully imitates demonstrations that would lead to novel states. Ecoffet et al. [16] also shows the benefit of exploration by returning to promising states. Our method can be viewed in general as an extension of [16], though we do not need to rely on the assumption that the environment is resettable to arbitrary states. Similar to previous off-policy methods, we use experience replay to enhance exploration. Many off-policy methods [25, 36, 1] tend to discard old experiences with low rewards and hence may prematurely converge to sub-optimal behaviors, but DTSIL using these diverse experiences has a better chance of finding higher rewards in the long term. Contemporaneous works [5, 4] as off-policy methods also achieved strong results on Atari games. NGU [5] constructs an episodic memory-based intrinsic reward using k-nearest neighbors over the agent's recent experience to train the directed exploratory policies. Agent57 [4] parameterizes a family of policies ranging from very exploratory to purely exploitative and proposes an adaptive mechanism to choose which policy to prioritize throughout the training process. While these methods require a large number of interactions, ours perform competitively well on the hard-exploration Atari games with less than one-tenth of samples. Model-based reinforcement learning [24, 47, 26] generally improves the efficiency of policy learning. However, in the long-horizon, sparse-reward tasks, it is rare to collect precious transitions with non-zero rewards and thus it is difficult to learn a model correctly predicting the dynamics of getting positive rewards. We instead perform efficient policy learning in the hard-exploration tasks because of efficient exploration with the trajectory-conditioned policy.

# 4 Experiments

In the experiments, we aim to answer the following questions: (1) How well does the trajectory-conditioned policy imitate the diverse demonstration trajectories? (2) Does the imitation of the past diverse experience enable the agent to further explore more diverse directions and guide the exploration to find the trajectory with a near-optimal total reward? (3) Is our method helpful for avoiding myopic behaviors and converging to near-optimal solutions?

We compare our method with the following baselines: (1) PPO [48]; (2) PPO+EXP: PPO with reward $f(r_t) + \lambda/\sqrt{N(e_t)}$, where $\lambda/\sqrt{N(e_t)}$ is the count-based exploration bonus, $N(e)$ is the number of times the cluster which the state representation $e$ belongs to was visited during training and $\lambda$ is the hyper-parameter controlling the weight of exploration term; (3) PPO+SIL: PPO with Self-Imitation

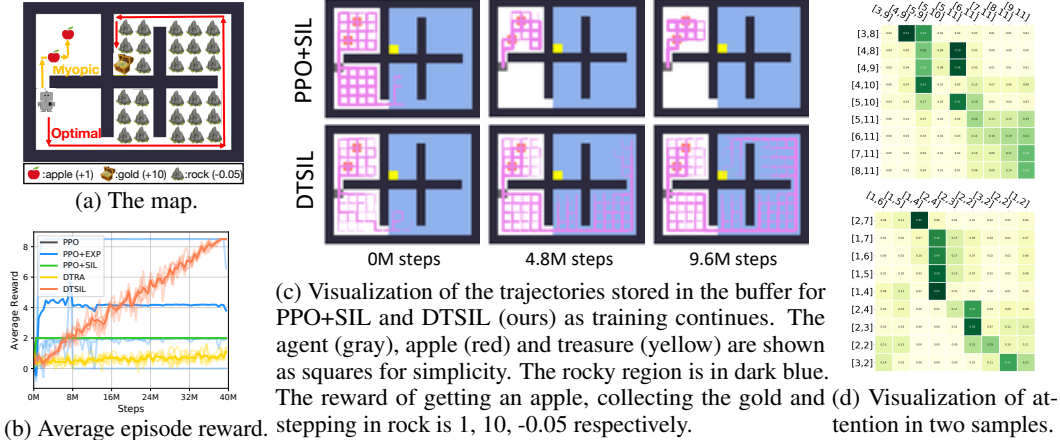

(a) The map.

(b) Average episode reward.

(c) Visualization of the trajectories stored in the buffer for PPO+SIL and DTSIL (ours) as training continues. The agent (gray), apple (red) and treasure (yellow) are shown as squares for simplicity. The rocky region is in dark blue. The reward of getting an apple, collecting the gold and stepping in rock is 1, 10, -0.05 respectively.

(d) Visualization of attention in two samples.

Figure 3: (a) The map of Apple-Gold domain. (b) Average reward of recent 40 episodes. The curves in dark colors are average over 5 curves in light colors. (c) Comparison of trajectories. (d) Attention in the learned trajectory-conditioned policy. The x-axis and y-axis correspond to the state (e.g. agent's location) in the source sequence (demonstration) and the generated sequence (agent's new trajectory), respectively. Each cell shows the weight $\alpha_{ij}$ of the $j$-th source state for the $i$-th target state.

Learning [36]; (4) DTRA ("Diverse Trajectory-conditioned Repeat Actions"): we keep a buffer of diverse trajectories and sample the demonstrations as DTSIL, but we simply repeat the action sequence in the demonstration trajectory and then perform random exploration until the episode terminates. More details about the implementation can be found in the appendix.

## 4.1 Apple-Gold Domain

The Apple-Gold domain (Fig. 3a) is a grid-world environment with misleading rewards that can lead the agent to local optima. At the start of each episode, the agent is placed randomly in the left bottom part of the maze. An observation consists of the agent's location $(x_t, y_t)$ and binary variables showing whether the agent has gotten the apples or the gold. A state is represented as the agent's location and the cumulative positive reward indicating the collected objects, i.e. $e_t = (x_t, y_t, \sum_{i=1}^{t} \max(r_i, 0))$.

In Fig. 3b, PPO+EXP achieves the average reward of 4. PPO+EXP agent can explore the environment and occasionally gather the gold to achieve the best episode reward around 8.5. However, it rarely encounters this optimal reward. Thus, this parametric approach might forget the good experience and fails to replicate the best past trajectory to achieve the optimal total reward. Fig. 3b shows that PPO+SIL agent is stuck with the sub-optimal policy of collecting the two apples with a total reward of 2 on average. Fig. 3c visualizes how the trajectories in the memory buffer evolve during the learning process. Obviously, PPO+SIL agent quickly exploits good experiences of collecting the apples and the buffer is filled with the trajectories in the nearby region. Therefore, the agent only adopts the myopic behavior and fails on this task. In the environment with the random initial location of the agent, repeating the previous action sequences is not sufficient to reach the goal states. The DTRA agent has a difficulty in exploring the environment and achieving good performance.

Unlike the baseline methods, DTSIL is able to obtain the near-optimal total reward of 8.5. Fig. 3c verifies that DTSIL can generate new trajectories visiting novel states, gradually expand the explored region in the environment, and discover the optimal behavior. A visualization of attention weight in Fig. 3d investigates which states in the demonstration are considered more important when generating the new trajectory. Even though the agent's random initial location is different from the demonstration, we can see a soft-alignment between the source sequence and the target sequence. The agent tends to pay more attention to states which are several steps away from its current state in the demonstration. Therefore, it is guided by these future states to determine the proper actions to imitate the demonstration.

## 4.2 Atari Games

We evaluate our method on the hard-exploration games in the Arcade Learning Environment [8, 30]. The environment setting is the same as [13]. There is a sticky action [30] resulting in stochasticity in the dynamics. The observation is a frame of raw pixel images, and the state representation $e_t = (\text{room}_t, x_t, y_t, \sum_{i=1}^{t} \max(r_i, 0))$ consists of the agent's ground truth location (obtained from

| Method | DTSIL+EXP | PPO+EXP | SmartHash | NGU* | Abstract-HRL | IDF | A2C+SIL | PPO+CoEX | RND | NGU | Agent57 |
|---|---|---|---|---|---|---|---|---|---|---|---|
| #Frames | 3.2B | 3.2B | 4B | 35B | 2B | 0.1B | 0.2B | 2B | 16B | 35B | 100B |
| Montezuma | **22,616** | 12,338 | 6,600 | **15,000** | 11,000 | 2,505 | 2,500 | 11,618 | 10,070 | 10,400 | 9,352 |
| Pitfall | **12,446** | 0 | - | - | 10,000 | - | - | - | -3 | 8,400 | **18,756** |
| Venture | **2,011** | 1,817 | - | - | - | 416 | 0 | 1,916 | 1,859 | 1,700 | **2,623** |

Table 1: Comparison with the state-of-the-art results. The top-2 scores for each game are in bold.Abstract-HRL [29] and NGU* (i.e., NGU with hand-crafted controllable states) [5] assume more high-level state information, including the agent's location, inventory, etc. DTSIL, PPO+EXP [13], and SmartHash [53] only make use of agent's location information from RAM. IDF [10], A2C+SIL [36], PPO+CoEX [13], RND [11], NGU [5] and Agent57 [4] (a contemporaneous work) do not use RAM information. The score is averaged over multiple runs, gathered from each paper, except PPO+EXP from our implementation.

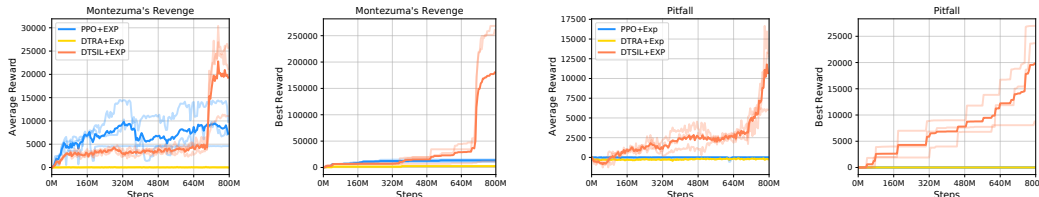

Figure 4: Learning curves of the average episode reward and the best episode reward found on Montezuma's Revenge and Pitfall, averaged over 3 runs. More statistics are reported in the appendix.

RAM) and the accumulated positive environment reward, which implicitly indicates the objects the agent has collected. It is worth noting that even with the ground-truth location of the agent, on the two infamously difficult games Montezuma's Revenge and Pitfall, it is highly non-trivial to explore efficiently and avoid local optima without relying on expert demonstrations or being able to reset to arbitrary states. Many complicated elements such as moving entities, traps, and the agent's inventory should be considered in decision-making process. Empirically, as summarized in Tab. 1, the previous SOTA baselines using the agent's ground truth location information even fail to achieve high scores.

Using the state representation $e_t$, we introduce a variant 'DTSIL+EXP' that adds a count-based exploration bonus $r_t^+ = 1/\sqrt{N(e_t)}$ to $r_t^{\mathrm{DTSIL}}$ for faster exploration.[1] DTSIL discovers novel states mostly by random exploration after the agent finishes imitating the demonstration. The pseudo-count bonus brings improvement over random exploration by explicitly encouraging the agent to visit novel states with less count. For a fair comparison, we also include count-based exploration bonus in DTRA. However, with stochasticity in the dynamics, it cannot avoid the dangerous obstacles and fails to reach the goal by just repeating the stored action sequence. Therefore, the performance of DTRA+EXP (Fig. 4) is poor compared to other methods.

On Venture (Tab. 1), it is relatively easy to explore and gather positive environment rewards. DTSIL performs only slightly better than the baselines. On Montezuma's Revenge (Fig. 4), in the early stage, the average episode reward of DTSIL+EXP is worse than PPO+EXP because our policy is trained to imitate diverse demonstrations rather than directly maximize the environment reward. Contrary to PPO+EXP, DTSIL is not eager to follow the myopic path (Fig. 1).[2] As training continues, DTSIL+EXP successfully discovers trajectories to pass the first level with a total reward more than 20,000. As we sample the best trajectories in the buffer as demonstrations, the average episode reward increases to surpass 20,000 in Fig. 4. On Pitfall, positive rewards are much sparser and most of the actions yield small negative rewards (time step penalty) that would discourage getting a high total reward in the long term. However, DTSIL+EXP stores trajectories with negative rewards, encourages the agent to visit these novel regions, discovers good paths with positive rewards and eventually attains an average episode reward over 0. In Fig. 4, different performances under different random seeds are due to huge positive rewards in some states on Montezuma's Revenge and Pitfall. Once the agent luckily finds these states in some runs, DTSIL can exploit them and perform much better than other runs.

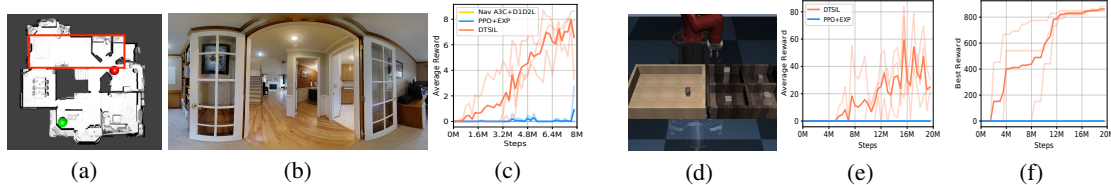

|     (a)     |     (b)     |     (c)     |     (d)     |     (e)     |     (f)     |

Figure 5: (a) Indoor scene for navigation task. (b) A panoramic view from a specific viewpoint. (c) Learning curves of average reward on navigation task. (d) Bin picking. (e) Learning curves of average reward on manipulation task. (f) Learning curves of best reward on manipulation task.

### 4.3 Continuous Control Tasks

When the initial condition is highly random, previous works imitating expert demonstrations (e.g. [3]) would also struggle. We slightly modify DTSIL to handle the highly random initial states: in each episode, from buffer $\mathcal{D}$, we sample the demonstration trajectory with a start state similar to the current episode. The detailed algorithm is described in the supplementary material.

**Navigation** We focus on a more realistic environment, a distant visual navigation task designed on Gibson dataset [55]. To make the task more challenging, the agent is randomly placed in the environment (red rectangle in Fig. 5a), a positive reward 10 is given only when it reaches a fixed target location (green point in Fig. 5a) which is significantly further away. The agent receives no information about the target (such as the target location or image) in advance. The observation is a first-person view RGB image and the state embedding is the agent's location and orientation (which is usually available in robotics navigation tasks) and the cumulative reward. This experiment setting is similar to the navigation task with a static goal defined in [31]. Apart from the baseline PPO+EXP, we also compare with Nav A3C+D1D2L [31], which uses the agent's location and RGB and depth image. This method performs well in navigation tasks on DeepMind Lab where apples with small positive rewards are randomly scattered to encourage exploration, but on our indoor navigation task, it fails to discover the distant goal without the exploration bonus. Fig. 5c shows that Nav A3C+D1D2L can never reach the target. PPO+EXP, as a parametric approach, is sample-inefficient and fails to quickly exploit the previous successful experiences. However, DTSIL agent can successfully explore to find the target and gradually imitate the best trajectories of reward 10 to replicate the good behavior.

**Manipulation** Bin picking is one of the hardest tasks in Surreal Robotics Suite [18]. Fig. 5d shows the bin picking task with a single object, where the goal is to pick up the cereal and place it into the left bottom bin. With carefully designed dense rewards (i.e. positive rewards at each step when the robot arm moving near the object, touching it, lifting it, hovering it over the correct bin, or successfully placing it), the PPO agent can pick up, move and drop the object [18]. We instead consider a more challenging scenario with sparse rewards. The reward is 0.5 at the single step of picking up the object, -0.5 if the object is dropped in the wrong place, 1 at each step when the object keeps in the correct bin. The observation is the physical state of the robot arm and the object. The state embedding consists of the position of the object and gripper, a variable about whether the gripper is open, and cumulative environment reward. Each episode terminates at 1000 steps. In Fig. 5f, PPO+EXP agent never discovers a successful trajectory with episode reward over 0, because the agent has difficulty in lifting the cereal and easily drops it by mistake. In contrast, DTSIL imitates the trajectories lifting the object, explores to move the cereal over the bins, finds trajectories successfully placing the object, exploits the high-rewarding trajectories, and obtains a higher average reward than the baseline (Fig. 5e).

### 4.4 Other Domains: Deep Sea and Mujoco Maze

In the supplementary material, we present additional details of the experimental results and also the experiments on other interesting domains. On **Deep Sea** [37], we show that the advantage of DTSIL becomes more obvious when the state space becomes larger and rewards become sparser. On **Mujoco Maze** [15, 33], we show that DTSIL helps avoid sub-optimal behavior in continuous action space.

## 5 Discussions: Robustness and Generalization of DTSIL

### 5.1 Robustness of DTSIL with Learned State Representations

Learning a good state representation is an important open question and extremely challenging especially for long-horizon, sparse-reward environments, but it is not the main focus of this work. However, we find that DTSIL can be combined with existing approaches of state representation learning if the high-level state embedding is not available. When the quality of the learned state

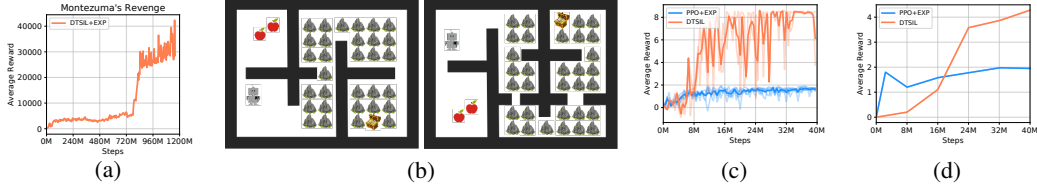

$$\begin{array}{cccc}
\text{(a)} & \text{(b)} & \text{(c)} & \text{(d)}
\end{array}$$

Figure 6: (a) Experiment with learned state representation. (b) Two samples of the random maze structure in Apple-Gold domain. (c) Learning curves of average episode reward on the training set of mazes in Apple-Gold domain. (d) Average episode reward on the test set of mazes for generalization experiment.

representation is not satisfactory (e.g., on Montezuma's Revenge, [13] fails to differentiate the dark rooms at the last floor), the trajectory-conditioned policy might be negatively influenced by the inaccuracy in $e_i^g$ or $e_i$. Thus, we modify DTSIL to handle this difficulty by feeding sequences of observations (instead of sequences of learned state embeddings) into the trajectory-conditioned policy. The learned state embeddings are merely used to cluster states when counting state visitation or determining whether to provide imitation reward $r^{im}$. Then DTSIL becomes more robust to possible errors in the learned embeddings. With the learned state representation from [13], on Montezuma's Revenge, DTSIL+EXP reaches the second level of the maze with a reward >20,000 (Fig 6a).

### 5.2  Robustness of DTSIL in Stochastic Environments

In the single-task RL problem, a Markov Decision Process (MDP) is defined by a state set $\mathcal{S}$, an action set $\mathcal{A}$, an initial state distribution $p(s_0)$, a state transition dynamics model $p(s_{t+1}|s_t, a_t)$, a reward function $r(s_t, a_t)$ and a discount factor $\gamma$. So the environment stochasticity falls into three categories: stochasticity in the initial state distribution, stochasticity in the transition function, and stochasticity in the reward function. For sparse-reward, long-horizon tasks, if the precious reward signals are unstable, the problem would be extremely difficult to solve. Thus, in this paper, we mainly focus on the other two categories of stochasticity. In Sec. 4.2 & 4.3, we show the efficiency and robustness of DTSIL in the environment with sticky action (i.e. stochasticity in $p(s_{t+1}|s_t, a_t)$) or highly random initial states (i.e. stochasticity in $p(s_0)$).

### 5.3  Generalization Ability of DTSIL

While many previous works about exploration focus on the single-task RL problem with a single MDP [53, 13, 16], we step further to extend DTSIL for the multiple MDPs, where every single task is in a stochastic environment with local optima. For example, in the Apple-Gold domain, we design 12 different structures of the maze as a training set (Fig. 6b). In each episode, the structure of maze is randomly sampled and the location of agent and gold is randomized in a small region. If the structure in the demonstration is different from the current episode, DTSIL agent might fail to recover the state of interest by roughly following the demonstration. Thus, using the buffer of diverse trajectories, we alternatively learn a hierarchical policy, which can behave with higher flexibility in the random mazes to reach the sampled states. We design the rewards so that the high-level policy is encouraged to propose the appropriate sub-goals (i.e., agent's locations) sequentially to maximize the environment reward and goal-achieving bonus (i.e. positive reward when the low-level policy successfully reaches the long-term goal sampled from the buffer). The low-level policy learns to visit sub-goals given the current observation (i.e. RGB image of the maze). The diverse trajectories in the buffer are also used with a supervised learning objective to improve policy learning. Fig. 6c shows that the hierarchical policy outperforms PPO+EXP during training. When evaluated on 6 unseen mazes in the test set, it can generalize the good behavior to some unseen environments (Fig. 6d). More details of the algorithm and experiments are in the supplementary material. Solving multi-task RL is a challenging open problem [35, 14, 44]. Here we verified this variant of DTSIL is promising and the high-level idea of DTSIL to leverage and augment diverse past trajectories can help exploration in this scenario. We leave the study of improving DTSIL furthermore as future work.

## 6  Conclusion

This paper proposes to learn diverse policies by imitating diverse trajectory-level demonstrations through count-based exploration over these trajectories. Imitation of diverse past trajectories can guide the agent to rarely visited states and encourages further exploration of novel states. We show that in a variety of stochastic environments with local optima, our method significantly improves count-based exploration method and self-imitation learning. It avoids prematurely converging to a myopic solution and learns a near-optimal behavior to achieve a high total reward.

## Broader Impact

DTSIL is likely to be useful in real-world RL applications, such as robotics-related tasks. Compared with previous exploration methods, DTSIL shows obvious advantages when the task requires reasoning over long-horizon and the feedback from environment is sparse. We believe RL researchers and practitioners can benefit from DTSIL to solve RL application problems requiring efficient exploration. Especially, DTSIL helps avoid the cost of collecting human demonstration and the manual engineering burden of designing complicated reward functions. Also, as we discussed in Sec. 5, when deployed for more problems in the future, DTSIL has a good potential to perform robustly and avoid local optima in various stochastic environments when combined with other state representation learning approaches.

DTSIL in its current form is applied to robotics tasks in the simulated environments. And it likely contributes to real robots in solving hard-exploration tasks in the future. Advanced techniques in robotics make it possible to eliminate repetitive, time-consuming, or dangerous tasks for human workers and might bring positive societal impacts. For example, the advancement in household robots will help reduce the cost for home care and benefit people with disability or older adults who needs personalized care for a long time. However, it might cause negative consequences such as large-scale job disruptions at the same time. Thus, proper public policy is required to reduce the social friction.

On the other hand, RL method without much reward shaping runs the risk of taking a step that is harmful for the environments. This generic issue faced by most RL methods is also applicable to DTSIL. To mitigate this issue, given any specific domain, one simple solution is to apply a constraint on the state space that we are interested to reach during exploration. DTSIL is complementary to the mechanisms to restrict the state space or action space. More principled way to ensure safety during exploration is a future work. In addition to AI safety, another common concern for most RL algorithms is the memory and computational cost. In the supplementary material we discuss how to control the size of the memory for DTSIL and report the cost. Empirically DTSIL provides ideas for solving various hard-exploration tasks with a reasonable computation cost.

## Acknowledgments and Disclosure of Funding

This work was supported in part by NSF grant IIS-1526059 and Korea Foundation for Advanced Studies. Any opinions, findings, conclusions, or recommendations expressed here are those of the authors and do not necessarily reflect the views of the sponsor.

## Footnotes

[1]The existing exploration methods listed in Table 1 take advantage of count-based exploration bonus (e.g., SmartHash, Abstract-HRL and PPO+CoEX). Therefore, a combination of DTSIL and the count-based exploration bonus does not introduce unfair advantages over other baselines.

[2]Demo videos of the learned policies for both PPO+EXP and DTSIL+EXP are available at https://sites.google.com/view/diverse-sil. In comparison to DTSIL+EXP, we can see the PPO+EXP agent does not explore enough to make best use of the tools (e.g. sword, key) collected in the game.

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
