[Supplementary Material]

# Memory Based Trajectory-conditioned Policies for Learning from Sparse Rewards

This supplementary material provides details of DTSIL algorithm in Appendix A & B. In Appendix C, G, and H, we clarify the details for the experiments. We compare DTSIL with the previous work in Appendix D. We also discuss how robust DTSIL is against the hyperparameter choices (Appendix E) and different types of stochasticity of the environment (Appendix C and F).

## Contents

# A   Detailed Description of DTSIL Algorithm

## A.1   DTSIL Full Algorithm

In Algorithm 1, we describe the full algorithm of our proposed method DTSIL.

---

**Algorithm 1** Diverse Self-Imitation Learning with Trajectory-Conditioned Policy (DTSIL)

---

Initialize parameter $\theta$ for the trajectory-conditioned policy $\pi_\theta(a_t | e_{\leq t}, o_t, g)$
Initialize the trajectory buffer $\mathcal{D} \leftarrow \emptyset$ **# Store diverse past trajectories**
Initialize set of transitions in the current episode $\mathcal{E} \leftarrow \emptyset$ **# Store current episode trajectory**
Initialize set of on-policy samples $\mathcal{F} \leftarrow \emptyset$ **# Store data for on-policy PPO update**
Initialize demonstration trajectory $g \leftarrow \emptyset$
**for** each iteration $i$ from 1 to $I$ **do**
   **for** each step $t$ **do**
      Observe $s_t = \{o_t, e_t\}$ and choose an action $a_t \sim \pi_\theta(a_t | e_{\leq t}, o_t, g)$
      Execute action $a_t$ in the environment to get $r_t, o_{t+1}, e_{t+1}$
      Store transition $\mathcal{E} \leftarrow \mathcal{E} \cup \{(o_t, e_t, a_t, r_t)\}$
      **# Positive reward if agent follows demonstration $g$**
      **# No reward after agent completes $g$ and then takes random exploration**
      Determine $r_t^{\text{DTSIL}}$ by comparing $e_{\leq t+1}$ with $g$      (Eq. 1)
      Store on-policy sample $\mathcal{F} \leftarrow \mathcal{F} \cup \{(o_t, e_t, a_t, g, r_t^{\text{DTSIL}})\}$
   **end for**
   **if** $s_{t+1}$ is terminal **then**
      $\mathcal{D} \leftarrow \text{UpdateBuffer}(\mathcal{D}, \mathcal{E})$      (Alg. 3)
      Clear current episode trajectory $\mathcal{E} \leftarrow \emptyset$
      $g \leftarrow \text{SampleDemo}(\mathcal{D}, i, I)$      (Alg. 2)
   **end if**
   $\theta \leftarrow \theta - \eta \nabla_\theta \mathcal{L}^{\text{RL}}$ **# Perform PPO update using on-policy samples**   (Eq. 2)
   Clear on-policy samples $\mathcal{F} \leftarrow \emptyset$
   $\theta \leftarrow \theta - \eta \nabla_\theta \mathcal{L}^{\text{SL}}$ **# Perform supervised learning updates using samples from $\mathcal{D}$ for $J$ times**   (Eq. 3)
**end for**

---

## A.2   Algorithm of Sampling Demonstrations

In Algorithm 2, we summarize how to sample the demonstrations from the trajectory buffer for exploration or exploitation. Considering the current iteration $i$ and the total number of iterations $I$, we set the probability of sampling demonstration for exploitation to learn good behavior as $\frac{i}{I}$ and the probability of sampling demonstration for exploration as $1 - \frac{i}{I}$.

---

**Algorithm 2** Sample Demonstration Trajectories

---

Input: the trajectory buffer $\mathcal{D} = \{e^{(1)}, \tau^{(1)}, n^{(1)}), (e^{(2)}, \tau^{(2)}, n^{(2)}), \cdots\}$
Input: current iteration $i$, total number of iterations $I$.
**# With probability $\frac{i}{I}$, run the exploitation mode; with probability $1 - \frac{i}{I}$, run the exploration mode**
**if** random number $\sim U[0, 1]$ is smaller than $\frac{i}{I}$ **then**
   **# sample one of the top-K trajectories reaching the near-optimal score in the buffer**
   $g \leftarrow \{e_0, e_1, \cdots, e_{|g|}\}$ for all $(o_t, e_t, a_t, r_t) \in \tau^{best}$
**else**
   Calculate probability distribution $p \leftarrow [\frac{1}{\sqrt{n^{(1)}}}, \frac{1}{\sqrt{n^{(2)}}}, \cdots]$
   $p \leftarrow \frac{p}{\sum_j p_j}$
   Sample $(e, \tau, n) \sim \text{Categorical}(\mathcal{D}, p)$
   $g \leftarrow \{e_0, e_1, \cdots, e_{|g|}\}$ for all $(o_t, e_t, a_t, r_t) \in \tau$
**end if**
**return** $g$

---

## A.3 Algorithm of Updating Trajectory Buffer

In Algorithm 3, we summarize how to process the newly collected episode and update the diverse trajectories in the trajectory buffer.

---

**Algorithm 3** Update Trajectory Buffer

---

Input: the trajectory buffer $\mathcal{D} = \{(e^{(1)}, \tau^{(1)}, n^{(1)}), (e^{(2)}, \tau^{(2)}, n^{(2)}), \cdots\}$
Input: the current episode $\mathcal{E} = \{(o_0, e_0, a_0, r_0), (o_1, e_1, a_1, r_1), \cdots, (o_T, e_T, a_T, r_T)\}$
Input: the threshold $\delta$ for high level state embedding
*# Consider all the states in $\mathcal{E}$*
**for** each step $t$ **do**
  *# Consider state $s_t$ and partial episode $\tau_{\leq t} = \{(o_0, e_0, a_0, r_0), \cdots, (o_t, e_t, a_t, r_t)\}$*
  **if** there exists $(e^{(k)}, \tau^{(k)}, n^{(k)}) \in \mathcal{D}$ where $\|e^{(k)} - e_t\| < \delta$ **then**
    *# Compare partial episode $\tau_{\leq t}$ with stored trajectory $\tau^{(k)}$*
    **if** $\tau_{\leq t}$ has higher total reward or reaches the same total reward with less steps **then**
      $\tau^{(k)} \leftarrow \tau_{\leq t} = \{(o_0, e_0, a_0, r_0), (o_1, e_1, a_1, r_1), \cdots, (o_t, e_t, a_t, r_t)\}$
      $e^{(k)} \leftarrow e_t$
    **end if**
    $n^{(k)} \leftarrow n^{(k)} + 1$
  **else**
    $\mathcal{D} \leftarrow \mathcal{D} \cup (e_t, \tau_{\leq t}, 1)$ where $\tau_{\leq t} = \{(o_0, e_0, a_0, r_0), (o_1, e_1, a_1, r_1), \cdots, (o_t, e_t, a_t, r_t)\}$
  **end if**
**end for**
**return** $\mathcal{D}$

---

## A.4 DTSIL Algorithm on Environments with Highly Random Initial States

DTSIL can successfully deal with the moderate degree of stochasticity such as the random initial delay and the sticky action on Atari games. In the environments with a highly random initial state distribution, such as the robotics navigation task where the agent is randomly placed in the house, we modify DTSIL algorithm to store the trajectories not only considering the ending state but also considering the start state. The modification of DTSIL method is summarized in Algorithm 5 and 4.

For each episode, we sample the demonstration with a start state similar to the current episode, so that the demonstration could appropriately lead the agent to the state of interest.

---

**Algorithm 4** Sample Demonstration Trajectories for DTSIL in Environments with Highly Random Initial State Distribution

---

Input: the trajectory buffer $\mathcal{D} = \{(e_{start}^{(1)}, e_{end}^{(1)}, \tau^{(1)}, n^{(1)}), (e_{start}^{(2)}, e_{end}^{(2)}, \tau^{(2)}, n^{(2)}), \cdots\}$
Input: current iteration $i$, total number of iterations $I$, the state state of the current episode $e$.

*# With probability $\frac{i}{I}$, run the exploitation mode; with probability $1 - \frac{i}{I}$, run the exploration mode*
**if** random number $\sim U[0,1]$ is smaller than $\frac{i}{I}$ **then**
   *# sample one of the top-K trajectories reaching the near-optimal score in the buffer, with a similar start state as $e$*
   $g \leftarrow \{e_0, e_1, \cdots, e_{|g|}\}$ for all $(o_t, e_t, a_t, r_t) \in \tau^{best}$ where $\|e_0 - e\| < \delta$
**else**
   *# sample one of the less frequently visited trajectories in the buffer, with a similar start state as $e$*
   Calculate probability distribution $p \leftarrow [\frac{1}{\sqrt{n^{(1)}}}, \frac{1}{\sqrt{n^{(2)}}}, \cdots]$
   **for** each stored trajectory $\tau^{(i)}$ **do**
      **if** $\|e_{start}^{(i)} - e\| \geq \delta$ **then**
         $p_i \leftarrow 0$
      **end if**
   **end for**
   $p \leftarrow \frac{p}{\sum_j p_j}$
   Sample $(e_{start}, e_{end}, \tau, n) \sim \text{Categorical}(\mathcal{D}, p)$
   $g \leftarrow \{e_0, e_1, \cdots, e_{|g|}\}$ for all $(o_t, e_t, a_t, r_t) \in \tau$
**end if**
**return** $g$

---

Figure 1: A diagram showing how to organize the buffer on environments with a highly random initial state distribution.

Additionally, we update the buffer considering trajectory concatenation to help exploration and exploitation of the good experiences. As illustrated in Figure 1, the agent occasionally collects a good trajectory from $e_0$ to $e_T$, where $e_T$ is a good state with positive reward. We can concatenate the stored trajectory from $e_{start}^{(j)}$ to $e_{end}^{(j)}$ and the trajectory from $e_t$ to $e_T$ if $e_t$ and $e_{end}^{(j)}$ are similar with each other. Therefore, the new concatenated trajectory $\tau$ (i.e. trajectory $e_{start}^{(j)}$ to $e_{end}^{(j)}$ and trajectory $e_t$ to $e_T$) could roughly lead the agent starting from $e_{start}^{(j)}$ to $e_T$ achieving positive reward.

---

**Algorithm 5** Update Trajectory Buffer for DTSIL in Environments with Highly Random Initial State Distribution

---

Input: the trajectory buffer $\mathcal{D} = \{(e_{start}^{(1)}, e_{end}^{(1)}, \tau^{(1)}, n^{(1)}), (e_{start}^{(2)}, e_{end}^{(2)}, \tau^{(2)}, n^{(2)}), \cdots \}$
Input: the current episode $\mathcal{E} = \{(o_0, e_0, a_0, r_0), (o_1, e_1, a_1, r_1), \cdots, (o_T, e_T, a_T, r_T)\}$
Input: the threshold $\delta$ for high level state embedding

*# Consider all the states in $\mathcal{E}$*
**for** each step $t$ **do**
   *# Consider state $s_t$ and partial episode $\tau_{\leq t} = \{(o_0, e_0, a_0, r_0), \cdots, (o_t, e_t, a_t, r_t)\}$*
   **if** there exists $(e_{start}^{(k)}, e_{end}^{(k)}, \tau^{(k)}, n^{(k)}) \in \mathcal{D}$ where $\|e_{start}^{(k)} - e_0\| < \delta$ and $\|e_{end}^{(k)} - e_t\| < \delta$ **then**
      *# Compare partial episode $\tau_{\leq t}$ with stored trajectory $\tau^{(k)}$*
      **if** $\tau_{\leq t}$ has higher total reward or reaches the same total reward with less steps **then**
         $\tau^{(k)} \leftarrow \tau_{\leq t} = \{(o_0, e_0, a_0, r_0), (o_1, e_1, a_1, r_1), \cdots, (o_t, e_t, a_t, r_t)\}$
         $e_{end}^{(k)} \leftarrow e_t$
         $e_{start}^{(k)} \leftarrow e_0$
      **end if**
      $n^{(k)} \leftarrow n^{(k)} + 1$
   **else**
      $\mathcal{D} \leftarrow \mathcal{D} \cup (e_0, e_t, \tau_{\leq t}, 1)$ where $\tau_{\leq t} = \{(o_0, e_0, a_0, r_0), (o_1, e_1, a_1, r_1), \cdots, (o_t, e_t, a_t, r_t)\}$
   **end if**
**end for**
*# If the current episode is with positive reward (which we rarely encounter in the long-horizon, sparse reward tasks), we consider concatenating the partial trajectory $\tau_{t:T}$ of the current episode with trajectories from other start states*
**if** $\sum_{i=0}^{T} r_i > 0$ **then**
   **for** each start state $e_{start}^{(j)}$ different from $e_0$ **do**
      **if** there exists $(e_{start}^{(j)}, e_{end}^{(j)}, \tau^{(j)}, n^{(j)}) \in \mathcal{D}$ where $\|e_{end}^{(j)} - e_t\| < \delta$ **then**
         *# Consider a new trajectory $\tau$ concatenating $\tau^{(j)}$ and $\tau_{t:T}$ with $e_{start} = e_{start}^{(j)}$ and $e_{end} = e_T$*
         **if** the new concatenated trajectory $\tau$ is novel or it's better that the stored trajectory with start state $e_{start}^{(j)}$ and end state $e_T$ **then**
            *add the concatenated trajectory $\tau$ into the buffer*
         **end if**
      **end if**
   **end for**
**end if**
**return** $\mathcal{D}$

---

Experiments on the robotics tasks in Section 4.3.3 in the main text and Appendix C.4 & C.5 show the advantage of our method over the baselines in such challenging environments.

Figure 2: An example showing the updates of $u$, given $\Delta t = 4$. At each step $t$, we check the state embedding $e_{t+1}$ to find similar state embedding $e_{u'}^g$ satisfying $e_{t+1} \approx e_{u'}^g$ (i.e. $\|e_{t+1} - e_{u'}^g\| < \delta$) and determine the reward according to Eq. 1. After completing the demonstration, the policy would perform random exploration until the episode terminates because the reward is always 0 ($e_6$).

## B  Additional Implementation Details

### B.1  Imitation Reward

Given a demonstration trajectory $g = \{e_0^g, e_1^g, \cdots, e_{|g|}^g\}$, we provide reward signals for imitating $g$ and train the policy to maximize rewards. At the beginning of an episode, the index $u$ of the lastly visited state embedding in the demonstration is initialized as $u = -1$. At each step $t$, if the agent's new state $s_{t+1}$ has an embedding $e_{t+1}$ and it is the similar enough to any of the next $\Delta t$ state embeddings starting from the last visited state embedding $e_u^g$ in the demonstration (i.e., $\|e_{t+1} - e_{u'}^g\| < \delta$ where $u < u' \le u + \Delta t$), then it receives a positive imitation reward $r^{\text{im}}$, and the index of the last visited state embedding in the demonstration is updated as $u \leftarrow u'$. This encourages the agent to visit the state embeddings in the demonstration in a soft-order so that the agent could possibly edit or augment the demonstration when executing a new trajectory. The demonstration plays a role to guide the agent to the region of interest in the state embedding space. In summary, the agent receives a reward $r_t^{\text{DTSIL}}$ defined as

$$r_t^{\text{DTSIL}} = \begin{cases} f(r_t) + r^{\text{im}} & \text{if } \exists u', u < u' \le u + \Delta t, \text{ such that } \|e_{u'}^g - e_{t+1}\| < \delta \\ 0 & \text{otherwise} \end{cases} \tag{1}$$

where $f(\cdot)$ is a monotonically increasing function (e.g., clipping [7]). Figure 2 illustrates the updates of $u$ during an episode when the agent visits a state whose embedding is close to state embeddings in the trajectory $g$.

To investigate how well the trajectory-conditioned policy imitates the diverse demonstrations, we define the success ratio as $\frac{u}{|g|}$ to measure the portion of demonstration imitated, where $u$ is the index of the last visited state embedding in $g$ when the agent's current episode terminates. We report the success ratio for the experiments in the following section C. On Apple-Gold domain, the average success ratio of the imitation increases as training goes on and eventually becomes close to 1.0, which indicates the trajectory-conditioned policy could successfully follow any given demonstration from the buffer.

### B.2  Trajectory-Conditioned Policy

In the trajectory-conditioned policy (Figure 3), we first encode the input state $e_t$ (or $e_i^g$) with a fully-connected layer with 64 units. Next, a RNN with gated recurrent units (GRU) computes the feature $h_t$ (or $h_i^g$) with 128 units. The attention weight $\alpha_t$ is calculated based on the Bahdanau attention mechanism [1]. The concatenation of the attention readout $c_t$, the hidden feature of agent's current state $h_t$, and the feature from the observation $\phi_t$ is used to predict $\pi(a_t|e_{\le t}, o_t, g)$ with a linear layer.

During training, our algorithm begins with an empty buffer $\mathcal{D}$. We initialize the demonstration as a list of zero vectors. With such an input demonstration, the agent performs the random exploration to collect trajectories to fill the buffer $\mathcal{D}$. In practice, when we accumulate more and more diverse trajectories in the buffer, the sampled demonstration trajectory $g = \{e_0^g, e_1^g, \cdots, e_{|g|}^g\}$ could be lengthy. We present a part of the demonstration as the input into the policy, similarly to translating

Figure 3: Architecture of the trajectory-conditioned policy (Repeat of Figure 2 Right in the main text.)

a paragraph sentence by sentence. Specifically, we first input $\{e_0^g, e_1^g, \cdots, e_m^g\}$ ($m \leq |g|$) into the policy. When the index of the agent's last visited state embedding in the demonstration $u$ belongs to $\{m - \Delta t + 1, \cdots, m\}$, we think that the agent has accomplished this part of the demonstration, and switch to the next part $\{e_u^g, e_{u+1}^g, \cdots, e_{u+m}^g\}$. We repeat this process until the last part of the demonstration. If the last part $\{e_u^g, e_{u+1}^g, \cdots, e_{|g|}^g\}$ is less than $m+1$ steps long, we pad the sequence with zero vectors.

## B.3    Buffer Organization

In DTSIL algorithm, we set a threshold $\delta$ to cluster similar state embeddings. In our experiments, the state embeddings mostly include the agent's location information and cumulative positive reward. If the $\ell_\infty$ distance between two state embeddings is smaller than the threshold $\delta$, we consider these two state embeddings are similar enough. Such a condition is considered when updating the buffer with a new trajectory and providing the imitation reward to encourage following the demonstration. Across different environments, the agent moves in the state embedding space with different ranges. On the Apple-Gold domain, the agent moves within a $17 \times 13$ discrete grid. On Atari games, the agent navigates around many rooms, and each room is shown in an $84 \times 84$ screen. On robotics navigation task, the agent may walk around a floor with coordinate $x, y, z$ in the ranges $[-7.5, 0.5] \times [-2, 2.5] \times [0, 1]$. To make it easier to set a proper value of $\delta$ for various environments, we normalize the ranges so that each dimension of the agent's location coordinate has a range $[0, 1]$, so we can set the hyper-parameter $\delta$ around 0.1 for all environments. The specific value of $\delta$ for each experiment is described in the following sections and the ablative study about $\delta$ is discussed in Appendix E.2.

We do not limit the size of buffer $\mathcal{D}$, but the memory cost of DTSIL algorithm is still controllable in practice. For each entry $(e^{(i)}, \tau^{(i)}, n^{(i)}) \in \mathcal{D}$, the most memory consuming part is $\tau^{(i)} = \{(o_0, e_0, a_0, r_0), (o_1, e_1, a_1, r_1), \cdots\}$. We notice that in the stored trajectory $\tau^{(i)}$, the state embedding $e_t$ (e.g. agent's location and cumulative reward) only needs small memory size, while the raw observation $o_t$ (e.g. RGB screen frame) is more expensive to store. Fortunately, $o_t$ in the demonstration trajectory is only useful in the supervised learning part. Therefore, we only store the $o_t$ in trajectory $\tau^{(i)}$, if the ending state $e^{(i)}$ is rarely visited. In such cases, supervised learning is greatly helpful to train the agent to imitate the demonstration trajectory $\tau^{(i)}$ and explore around the rarely visited state $e^{(i)}$. Otherwise, we do not store $o_t$ in the trajectory $\tau^{(i)}$, if the ending state $e^{(i)}$ has a high number of visitation count. The state $e^{(i)}$ could be frequently visited by the policy, so we do not need to leverage supervised learning objective to push the agent to $e^{(i)}$ and thus we can save the memory by not storing $o_t$ in the trajectory $\tau^{(i)}$. We report the memory cost for each of our experiments in the following sections.

## C   Additional Experimental Details

### C.1   Apple-Gold Domain with Random Initial States

The Apple-Gold domain is a simple maze with a 17x13 grid. The observation at each step $t$ is the agent's location $(x_t, y_t)$ and binary variables indicating whether the apple or gold has been collected. The action space is discrete with 5 possible actions: up, down, left, right and noop. The reward of getting an apple, collecting the gold and taking a step in the rocky region is +1, +10, -0.05 respectively. The high-level state representation $(x_t, y_t, \sum_{i=1}^{t} \max(0, r_i))$ is the agent's location and cumulative positive reward at step $t$, indicating the collected objects. At the start of each episode, the agent is randomly placed in the left bottom part in the maze, i.e. the pink rectangle in Figure 4a. The time limit for each episode is 45 steps. An episode terminates when the agent reaches the time limit or finds the gold. Therefore, the optimal path within the time limit is to walk through the rocky region and collect the gold, achieving the highest episode reward 8.5.

For experiments on the Apple-Gold domain, we simplify the architecture of our trajectory-conditioned policy, i.e. the features from $o_t$ are not used for the policy. We set the reward function $f(r_t) = r_t$, learning rate $\eta = 2.5e - 4$, length of demonstration input into the policy $m = 10$, $\Delta t = 10$, threshold for clustering the state embedding $\delta = 0.1$ after normalizing the range of the agent's location coordinate. The number of supervised learning update $J = 10$ if the action accuracy in supervised learning is less than 0.75. Otherwise, the number of supervised learning update $J = 1$. In exploitation mode, we imitate the top-1 best trajectory with the highest total reward. In this experiment setting, there could only be a single trajectory avoiding the myopic behavior and getting the optimal episode reward in the buffer. If we imitate top-K best trajectories ($K > 1$), it's impossible to train an agent to always walk towards the gold.

For the baseline method PPO+SIL, we search the hyper-parameter $M$ (number of SIL updates for each PPO update) among $\{0.5, 1, 2, 4\}$, and show the best result we achieved. For the baseline method PPO+EXP, we search the weight of count-based exploration bonus $\lambda \in \{5, 10, 20, 50\}$ and report the best result we achieved. For the baseline method DTRA, there is not policy learning. We use the same hyper-parameters as DTSIL to update the buffer and sample demonstrations.

|         |         |         |         |         |
|---------|---------|---------|---------|---------|
| (a) The map | (b) Average reward | (c) Best reward | (d) Success ratio | (e) Number of state |

Figure 4: (a) The map of Apple-Gold domain. (b)-(e) Learning curves of the average episode reward, the best episode reward, the average success ratio and the number of different states found, where the curves in dark colors are average over 5 curves in light colors. The x-axis and y-axis correspond to the number of steps and statistics about the performance, respectively. The average reward and average imitation success ratio are the mean values over 40 recent episodes. The number of found state is the number of entries we collected in the buffer, where each entry represents a cluster of states in the state embedding space.

Figure 4 summarized our results comparing DTSIL with baseline methods. As we discussed in Section 4.1 in the main text, DTSIL outperforms the baselines obtaining higher average reward and discovering more states in the environment. Empirically, for DTSIL algorithm, there are around 400 different trajectories stored in the buffer and the memory cost is about 10MB.

### C.2   Atari Montezuma's Revenge with Random Initial Delay

On the Atari games, we set the environment MontezumaRevengeNoFrameskip-v4 with a random number (between 0 and 30) of noop action before each episode starts [7]. The observation $o_t$ is the last four gray frames stacked. So the observation has shape $84 \times 84 \times 4$. The action space is discrete with 18 possible actions. On Montezuma's Revenge, the reward is mostly 0 and it's positive when the agent collects objects. The high-level state representation at step $t$ is $(\mathrm{room}_t, x_t, y_t, \sum_{i=1}^{t} \max(r_i, 0))$ where $(x_t, y_t)$ is in $84 \times 84$ grid and the cumulative positive reward indicates the objects have been collected by the agent. The time limit for one episode is 4500 steps.

On the Atari games, with DTSIL algorithm, it is necessary to take the raw observation $o_t$ as input into the trajectory-conditioned policy because the location information in $e_{\leq t}$ solely could not make the agent take temporal context into account (e.g. avoiding moving skulls and passing laser gates). We input the raw observation $o_t$ with shape $84 \times 84 \times 4$. Three convolutional layers are used to encode $o_t$ and then the convolutional feature $\phi_t$ is flattened and concatenated with other features.

The reward function is $f(r_t) = 2 \cdot \text{clip}(r_t, 0, 1)$. The learning rate is $2.5e - 4$. The length of demonstration input into the policy is $m = 20$. $\Delta t$ is 20. The threshold for clustering the state embedding is $\delta = 0.1$, so the state embedding in each room is roughly divided into 10x10 clusters if the cumulative reward is the same and 10x10 discretization is used in previous work [3]. The number of supervised learning update $J = 10$ if the action accuracy in supervised learning is less than 0.75. Otherwise, the number of supervised learning update $J = 1$. In exploitation mode, we imitate the top-10 best trajectories with the highest total rewards.

For the baseline method PPO+EXP, we search the weight of count-based exploration bonus $\lambda$ among $\{0.5, 1, 2, 4\}$ and report the best average reward we achieved. Figure 5 shows the advantage of DTSIL over the baselines. The buffer on Montezuma's Revenge costs around 6GB of memory.

Figure 5: Learning curves of the average episode reward, the best episode reward, the number of different rooms and the number of different states found on Montezuma's Revenge and Pitfall, where the curves in dark colors are average over 3 curves in light colors. The x-axis and y-axis correspond to the number of steps and statistics about the performance, respectively. The average reward is the mean values over 40 recent episodes. On Montezuma's Revenge, DTSIL+EXP discovers around 40 rooms on average, and it usually clears the first level and move on to the next level. In contrast, PPO+EXP never finds a path to clear the first level.

## C.3 Atari with Sticky Action

Figure 6: Learning curves of the average episode reward, the best episode reward, the number of different rooms and the number of different states found on Montezuma's Revenge Pitfall, and Venture. where the curves in dark colors are average over 3 curves in light colors. The x-axis and y-axis correspond to the number of steps and statistics about the performance, respectively. The average reward is the mean values over 40 recent episodes. On Montezuma's Revenge, DTSIL+EXP discovers 24 rooms on average. So it clears the first level. In contrast, PPO+EXP never finds a path to clear the first level.

We further set the environment on Atari games with the sticky action [6]. At each step, the agent randomly takes the previous action with a probability of 0.25. The other details for environment setup and hyper-parameters are the same as Appendix C.2. Figure 6 shows the performance of DTSIL and baselines in this setup. On Montezuma's Revenge and Pitfall, the environment rewards are super sparse and misleading rewards tend to trap the agent in the sub-optimal behavior. However, DTSIL+EXP has a higher chance to escape from the sub-optimal solutions and significantly outperforms the baseline approaches. Because of the stochasticity in the dynamic, imitation of diverse demonstrations is more difficult than in the Atari environment with random initial delay. We only take the exploration mode for the first 700M steps to increase the chance of finding the good trajectories going through the first level. Then in the last 100M steps, we take exploitation mode to imitate the best trajectories found during training to maximize average episode rewards. The buffer on Montezuma's Revenge, Pitfall and Venture costs the memory of 6GB, 500MB, 6GB respectively.

## C.4   Robotics Navigation Task with Highly Random Initial States

(a) The map    (b) Average reward    (c) Best reward    (d) Number of states    (e) Success ratio

Figure 7: (a) The map of Beechwood layout from Gibson dataset, where the initial location of the agent is in the red rectangle. (b)-(e) Learning curves of the average episode reward, the best episode reward, the number of different states and the average success ratio, where the curves in dark colors are average over 3 curves in light colors. The x-axis and y-axis correspond to the number of steps and statistics about the performance, respectively. The average reward and average imitation success ratio are the mean values over 40 recent episodes.

In the robotics navigation task, the agent receives observations as RGB frames from the first-person view. The action space is discrete with four possible actions: move forward, move backward, turn left and turn right. There is only positive reward 10 when the agent reaches the fixed target location. The state embedding is the agent's location and orientation $(x_t, y_t, z_t, \mathrm{roll}_t, \mathrm{pitch}_t, \mathrm{yaw}_t)$. The time limit for each episode is 250 steps.

The trajectory-conditioned policy takes the input of raw observation $o_t$ with shape $128 \times 128 \times 3$. Three convolutional layers are used to encode $o_t$ and then the convolutional feature $\phi_t$ is flattened and concatenated with other features. The reward function is $f(r_t) = r_t$. The learning rate is $2.5e - 4$. The length of demonstration input into the policy is $m = 10$. $\Delta t$ is 10. The threshold for clustering the state embedding is $\delta = 0.1$ in the normalized state embedding space. The number of supervised learning update $J = 10$ if the action accuracy in supervised learning is less than 0.75. Otherwise, the number of supervised learning update $J = 1$. In exploitation mode, we imitate the top-10 best trajectories with the highest total rewards.

For the baseline method PPO+EXP, we search the weight of count-based exploration bonus $\lambda$ among $\{5, 10, 20, 50\}$ and report the best average reward we achieved. DTSIL achieves higher average reward from the environments. Figure 7 (b)(c) is the same as Figure 10 in the main text. We report the number of states and success ratio here to provide more information about the training process. As shown in Figure 7, there are about 1000 diverse trajectories ending with diverse states stored in the buffer, and the memory usage is around 5GB.

## C.5   Robotics Manipulation Task with Highly Random Initial States

In the robotics manipulation task (bin picking, as shown in Figure 8a), we randomly set the initial location of the object in a left square and the agent needs to pick up the cereal and move it to the left bottom bin in the right part. At each step, the agent receives an observation as a 44–dim vector, consisting of the physical features of the robot arm and the position of the object. The high-level state embedding includes the information about the position of the robot arm $(x_t^{\mathrm{rbt}}, y_t^{\mathrm{rbt}}, z_t^{\mathrm{rbt}})$, the position of the object $(x_t^{\mathrm{obj}}, y_t^{\mathrm{obj}}, z_t^{\mathrm{obj}})$, whether the gripper is open or close $s_t^{\mathrm{grip}}$, and the cumulative environment reward $\sum_{i=0}^{t} r_i$. The discrete action space includes 7 possible actions, moving right,

(a) Bin Picking    (b) Average reward    (c) Best reward    (d) Number of states    (e) Success ratio

Figure 8: (a) A snapshot of Bin Picking task. (b)-(e) Learning curves of the average episode reward, the best episode reward, the number of different states and the average success ratio, where the curves in dark colors are average over 3 curves in light colors. The x-axis and y-axis correspond to the number of steps and statistics about the performance, respectively. The average reward and average imitation success ratio are the mean values over 40 recent episodes.

left, up, down, forward or backward and opening/closing the gripper. The agent will receive $0.5$ reward each time it picks up the cereal and $-0.5$ reward each time it releases it. The agent will receive a reward of $1$ every timestep as long as the cereal is in the correct bin. The time limit for each episode is 1000 steps.

The learning rate is $2.5e - 4$. The length of demonstration input into the policy is $m = 40$ and $\Delta t = 40$. We provide the information in the demonstration and the imitation reward more generously for this task because it is challenging to stably lift the object and it requires careful manipulation of the gripper and adjustment of the actions according to many influence factors such as the angle, velocity, friction force, etc. The threshold for clustering the state embedding is $\delta = 0.1$. The number of supervised learning update $J = 10$ if the action accuracy in supervised learning is less than 0.75. Otherwise, the number of supervised learning update $J = 1$. In exploitation mode, we imitate the top-10 best trajectories with the highest total rewards. The high-reward trajectories spend around 150 steps out of 1000 steps to successfully move the object into the correct bin. When the agent is exploiting these best trajectories, the agent successfully imitates the portion to place the object in the correct bin. But the final location of the object might not be quite close to the demonstration (even if both these two are in the correct bin). So the episode success ratio is finally around 0.15. During training, we store around 2000 trajectories in total and the whole replay buffer costs around 120MB of memory.

## C.6 Deep Sea

Figure 9: Deep Sea exploration: a simple example where deep exploration is critical.

As introduced in [8], the Deep Sea environment consists of a $N \times N$ grid where a state is represented as a one-hot encoding (i.e., tabular states). The agent begins each episode in the top left corner of the grid and descends one row per timestep. Each episode terminates after N steps when the agent reaches the bottom row. In each state, there is a random but fixed mapping between actions $A = \{0, 1\}$ and the transitions 'left' and 'right'. At each timestep there is a small cost $r = -0.01/N$ of moving right, and $r = 0$ for moving left. However, should the agent transition right at every timestep of the episode it will be rewarded with an additional reward of +1. This presents a particularly challenging

exploration problem for two reasons. First, following the 'gradient' of small intermediate rewards leads the agent away from the optimal policy. Second, a policy that explores with actions uniformly at random has probability $2^{-N}$ of reaching the rewarding state in any episode.

Figure 10: Experiment on Deep Sea. The learning curves show the average episode reward, best episode reward, the number of found state representations, and the average success ratio of imitating the demonstrations in order. The curves are averaged over 5 independent runs.

We compare DTSIL and baselines on deep sea environments with $10 \times 10$ grid and and $30 \times 30$ grid. The state embedding we use here is exactly the observation. The result is shown in Figure 10. In the first environment, it is easy for all of the methods to converge to optimal behavior. The second one is much more challenging to find the optimal trajectory maximizing total reward. Therefore, PPO and PPO+SIL fail in such a environment due to hard exploration. PPO+EXP is unable to always explore to find good behavior and exploit it efficiently within 12M timesteps. DTSIL successfully discovers the right way and imitate to converge to the optimal behavior.

## C.7 Mujoco Maze

We evaluate DTSIL on continuous control tasks. We adapt the maze environment introduced in [4] to construct a set of challenging tasks, which requires the point mass agent to collect the key, open the door with the same color as the key and finally reach the treasure to get a high score. One key cannot be re-used once it was used before to open a door with the same color. This makes the agent to be easily trapped at sub-optimal behavior. A visualization of these environments is shown in Figure 11. The agent's initial location is randomly sampled from a Gaussian distribution as in standard MuJoco tasks [2]. The observation is the agent's location and range sensor reading about nearby objects. The state representation is $e_t = (x_t, y_t, \sum_{i=1}^{t} r_i)$.

Figure 11: Point Maze in Mujoco domain. The reward for getting the key, opening the door, and collecting the treasure (yellow block) is 1, 2, and 6 respectively. The learning curve of the episode reward is averaged over 3 independent runs.

As shown in the first maze of Figure 11, the agent can easily get the blue key near its initial location and open the blue door in the upper part. However, the optimal path is to bring the key to open the blue door in the bottom and obtain the treasure, reaching an episode reward of 9. In the second maze, the agent should bring the blue key and pick up the green key while avoiding opening the blue door in the upper part. Then, the green and blue key can open the two doors at the bottom of the maze, which results in the total reward of 12. The learning curves in Figure 11 show that PPO, PPO+EXP, and PPO+SIL may get stuck at a sub-optimal behavior, whereas our policy eventually converges to the behavior achieving the high episode reward.

## C.8 Montezuma's Revenge with Learned State Representation

We learn a state representation with the approach proposed in [3]. In the state embedding $(\text{room}_t, x_t, y_t, \sum_{i=1}^{t} \max(r_t, 0))$, $(x_t, y_t)$ can be detected by a well-trained ADM, and the room information is based on the clustering of projected features of the observation. However, on Montezuma's Revenge, the rooms at the bottom floor are all black and it's impossible to differentiate them based on visual features if the agent does not pick up a torch to lit up the rooms. So the learned state representation is not reliable when the agent goes to the last floor. We modify DTSIL to make it more robust against the inaccuracy in the learned state embeddings. In the trajectory-conditioned policy, instead of using the sequences of learned state representations as input, we provide the sequence of observations in the demonstration and the agent's incomplete trajectory. Then the learned state representations are mainly used to cluster the states, organize the trajectories in the buffer and assign imitation rewards. With this small modification, on Montezuma's Revenge with random initial delay, DTSIL performs robustly with the learned state representation, eventually achieves an average score over 20,000 and visits the next level. In conclusion, we could combine DTSIL with other methods of state representation learning to remove our assumption about the availability of high-level state embedding.

Figure 12: Learning curves of the average reward, the best episode reward, the number of rooms, the number of different state representations found on Atari Montezuma's Revenge.

## C.9 Apple-Gold domain with Highly Random Structure

We further investigate the efficiency of DTSIL for multi-task problems, where each task is defined in a stochastic environment with local optima. On the Apple-Gold domain, we design 12 possible structures of mazes as shown in Figure 14. For each episode, the structure of the maze is randomly sampled and the initial location of the agent and the location of the gold is randomized. For exploration and exploitation, we sample the state of interest from the buffer of diverse trajectories. We learn a hierarchical policy with the buffer so that the agent could behave flexibly in the highly stochastic environment to reach the sampled long-term goal.

(a) A diagram for the high-level policy.      (b) A diagram for the low-level policy.

Figure 13: A diagram for the hierarchical policy we learn on the Apple-Gold domain.

As shown in Figure 13b, at every step $t$, the low-level policy observes $o_t$ (e.g. a RGB image of the map on Apple-Gold domain) and a sub-goal $g_t$ (e.g. a gray image indicating the target location of the agent) proposed by the high-level policy and output an atomic action $a_t$. For every $c$ steps, in Figure 13a, the high-level policy observes $o_t$ and produces a high-level action to update the sub-goal $e_t^g$ (e.g. the target location of the agent) conditioning on the long-term goal state $g$ sampled from the buffer. The high-level policy receives the environment rewards $r_t$ and the goal-achieving bonus (e.g. positive reward when the low-level policy successfully reaches the goal location $e^g$) while the low-level policy only gets a positive reward when visiting the sub-goals $e_t^g$. To improve policy learning and to better leverage the past trajectories, we introduce the supervised learning objective. We sample a trajectory $\tau = \{(o_0, e_0, a_0, r_0), (o_1, e_1, a_1, r_1) \cdots\} \in \mathcal{D}$, formulate the long-term goal state $g = o_{|g|}$, and assume the agent's current observation is $o_t$ for $1 \leq t \leq |g|$. In such a case, the 'correct' action for the high-level policy is to propose the location $e_{t+c}$ as the sub-goal and the correct action for the low-level policy is $a_t$. Our supervised learning objective is to maximize the log probability of taking such actions. Experimental results in Figure 16 shows the performance of the hierarchical policy with the diverse trajectories. As we learn the hierarchical policy, we also evaluate it on the test set (6 structures unseen during training as shown in Figure 15). DTSIL outperforms the baseline PPO+EXP on the training set and test set.

Figure 14: Visualization of the maze structures in the training set. The agent (gray), apple (red), gold (yellow) are shown as squares for simplicity. The rocky region is in dark blue. On each maze, the initial location of the agent and the location of the gold could be randomized in a small region.

Figure 15: Visualization of the maze structures in the test set.

## D   Comparison with Learning Diverse Policies by SVPG

We replicate the method in [5] to learn diverse policies with the Stein variational policy gradient (SVPG). Their experiments focus on continuous control tasks with relatively simple observation

(a) Average episode reward    (b) Best episode reward    (c) Number of found states    (d) Average episode reward

Figure 16: (a)-(c) Learning curves of the average episode reward, the best episode reward, the number of different states where the curves in dark colors are average over 3 curves in light colors. The x-axis and y-axis correspond to the number of steps and statistics about the performance, respectively. The average reward is the mean values over 40 recent episodes. (d) Average episode reward on the test set.

spaces with limited local optimal branches in the state space. We learn 8 diverse policies in parallel following their method on our Apple-Gold domain with discrete action space. Figure 17 shows a visualization of the learning progress: the 8 policies learn to cover different regions of the environment. The SVPG method explores better than PPO+SIL, but the exploration of each individual agent is not strong enough to find the optimal path to achieve the highest episode reward.

Figure 17: Visualization of the trajectories stored in the buffer for PPO+SIL, SVPG diverse [5] and our method as training continues. In the second row, we show the trajectories for a total of 8 policies learned simultaneously with the SVPG method proposed in [5], where each color corresponds to the trajectories collected by each policy.

# E    Effects of Hyperparameters

## E.1    Hyperparameter $\Delta t$

$\Delta t$ influences how flexibly the demonstration should be followed. If $\Delta t = 1$, the agent could only get imitation reward $r^{im}$ when it visits the next state from the lastly visited state in the demonstration trajectory, i.e. $e_{u+1}^g$, where $u$ is the index of last visited state in the demonstration. With a larger value of $\Delta t$, we provide imitation reward $r^{im}$ if the agent visits any of the next $\Delta t$ state in the demonstration, i.e. any state in the set $\{e_{u+1}^g, e_{u+2}^g, \cdots, e_{u+\Delta t}^g\}$. In our experiment, we have the constraint $\Delta t \leq m$, where $m$ is the length of the input demonstration part. For each step, only the next $m$ states from the lastly visited state in the demonstration might be input into the trajectory-conditioned policy. So we should only consider awarding the agent for imitation if it visits a state in the demonstration that it has

known from the policy input. We run experiments on the Apple-Gold domain, Montezuma's Revenge,

Figure 18: Learning curves of the average reward.

and Pitfall with $m = 10$ and different values of $\Delta t$, as shown in Figure 18. On the Apple-Gold domain with a simple action space and observation space, it's easy to imitate the demonstration trajectories. Thus, DTSIL performs well for all values of $\Delta t \in \{2, 4, 6, 8, 10\}$. On Montezuma's Revenge and Pitfall, when the demonstration is longer and harder to follow, larger $\Delta t$'s are more suitable in order to provide imitation rewards more leniently. Therefore, we prefer to set a large value of $\Delta t$ to generously provide imitation rewards, and we set $\Delta t = m$ in our experiments.

### E.2   Hyperparameter $\delta$

$\delta$ is the threshold to cluster similar state embeddings. If $\delta$ is small, there could be a large number of clusters in the state embedding space, and it costs more trials to search for novel states with higher rewards. If $\delta$ is large, some distinctive state embeddings might be mistakenly clustered into one single cluster, and we may miss a chance to explore around a specific state embedding because in the buffer it can be replaced by another state embedding meaningfully different from itself.

Figure 19: Learning curves of the average reward.

Figure 19 shows the performance of DTSIL with different values of $\delta$. Based on the ablative study, we conclude that DTSIL is not quite sensitive to the choice of hyper-parameter $\delta$. After normalizing the state embedding space so that each dimension for the agent's location has a range $[0, 1]$ (as we discussed in Appendix C), $\delta \in [0.08, 0.14]$ is proper across different environments. With such a value of $\delta$, the number of clusters in the state embedding space is reasonable for exploration. The similar states are clustered into a single cluster, but meaningfully different states are clustered into different clusters. Thus, DTSIL could perform well with the proper hyper-parameter $\delta$.

Assume agent's location in state embeddings is normalized to [0,1] for each coordinate and the distance metric is $l_\infty$. When clustering embeddings in parametric memory, $\delta = 0.1$ will discretize 2D location space into $10 \times 10$ grid, an intuitively reasonable size. So we set $\delta = 0.1$ in our experiments.

# F   Effect of Stochasticity in Environments

Figure 20: Effects of the degree of stochasticity on DTRA and DTSIL. The imitation success ratio and average episode reward is averaged over the last 40 episodes during training.

On the Apple-Gold domain, we study the performance of DTSIL and DTRA when the agent takes a sticky action (e.g. the agent randomly repeats the previous action with a probability $p_{sticky}$). Figure 20 summarizes the final performance of DTRA and DTSIL on the Apple-Gold domain with different degrees of stochasticity in the dynamics. If $p_{sticky} = 0$, the dynamics are totally deterministic, and hence the DTRA agent repeating the stored action sequence could exactly go to the goal state. DTRA achieves the imitation success ratio 1, which indicates that the agent successfully follows the demonstration trajectory to the last state. However, even if the DTRA agent perfectly follows the diverse demonstrations, after visiting the goal state, it uniformly samples the action from the discrete action for each step until the episode terminates. Such a naive random exploration without the learned parameterized policy is not efficient enough to discover the optimal path with 40M training timesteps. On the contrary, DTSIL could gradually learn to imitate the diverse demonstrations, increase the imitation success ratio to 1, and achieve the optimal total reward. If $p_{sticky} > 0$, the performance of DTRA becomes much worse because it could not follow the demonstration well by repeating the stored action sequence in the stochastic environment. However, DTSIL is robust to the different degrees of sticky action in the dynamics. Figure 21, 22, 23, 24 shows the learning curves during training process.

Figure 21: $p_{sticky} = 0$

Figure 22: $p_{sticky} = 0.1$

Figure 23: $p_{sticky} = 0.2$

Figure 24: $p_{sticky} = 0.3$

The commonly used value for sticky action is 0.25 [6]. We present the experimental results with $p_{sticky} = 0.25$ in Figure 25.

(a) Average reward     (b) Best reward     (c) Success ratio     (d) Number of state

Figure 25: Learning curves of the average episode reward, the best episode reward, the average success ratio and the number of different states found, where the curves in dark colors are average over 5 curves in light colors.

# G  Hyperparameters

The hyperparameters used in each experiment are listed in Table 1. The robotics manipulation task requires careful adjustment of the action according to many factors such as the angle, velocity, friction force, etc. To provide the agent with more guidance signal, we input a longer part of the demonstration into the policy $m = 40$ and award the imitation reward more generously $\Delta t = 40$. On Mujoco maze, RL loss alone worked well so $J = 0$ we did not include SL loss for behavior cloning. On the other environments when action prediction in supervised learning is poor, we set a large $J$ to quickly learn to imitate demonstrations. When action prediction is accurate enough, we de-emphasize behavior cloning to enhance exploration around the demonstration. On Apple-Gold and Deep Sea environments, there could be only a single trajectory performing well enough to avoid the myopic behavior, so we only imitate the top-1 trajectory in exploitation mode. In the other environments, we imitate the top-10 best trajectories with the highest total rewards.

| Environment | Apple-Gold | Atari | Navigation | Deep Sea | Mujoco maze | Manipulation |
|---|---|---|---|---|---|---|
| Learning Rate $\eta$ | 2.5e-4 | 2.5e-4 | 2.5e-4 | 2.5e-4 | 1e-4 | 2.5e-4 |
| $\Delta t$ | 10 | 20 | 10 | 10 | 10 | 40 |
| Length of demonstration input part $m$ | 10 | 20 | 10 | 10 | 10 | 40 |
| Number of supervised learning updates $J$ | 10 if action prediction accuracy $\geq 0.75$; 1 otherwise | 10 if action prediction accuracy $\geq 0.75$; 1 otherwise | 10 if action prediction accuracy $\geq 0.75$; 1 otherwise | 10 if action prediction accuracy $\geq 0.75$; 1 otherwise | 0 | 10 if action prediction accuracy $\geq 0.75$; 1 otherwise |
| Threshold $\delta$ for state embedding | 0.1 | 0.1 | 0.1 | 0.1 | 0.1 | 0.1 |
| Top-$K$ trajectories imitation | 1 | 10 | 10 | 1 | 10 | 10 |
| Weight of exploration $\lambda$ in PPO+EXP | 10 (best one among 5, 10, 20, 50) | 1 (best one among 0.5, 1, 2, 4) | 10 (best one among 5, 10, 20, 50) | 0.2 (best one among 0.1, 0.2, 0.5, 1) | 1 (best one among 0.5, 1, 2, 4) | 10 (best one among 5, 10, 20, 50) |
| Discount factor $\gamma$ | 0.99 | 0.99 | 0.99 | 0.99 | 0.99 | 0.99 |

Table 1: Hyperparameters on various environments for our experiments.

# H  Environment Setting

For each experiment we conducted, we list the detailed environment setting in Table 2. There is stochasticity in the environments of the Apple-Gold domain, Atari, robotics tasks, and Mujoco maze, where the stochasticity lies in the initial state distribution or the dynamics.

| Environment | Apple-Gold | Atari | Navigation | Deep Sea | Mujoco maze | Manipulation |
|---|---|---|---|---|---|---|
| Observation | agent's location in 17x13 grid; binary variables indicating whether apple or gold is collected | stacked most recent 4 gray observations with shape 84x84x4 | first-view RGB frame with shape 84x84x3 | one-hot encoding of state in 10x10 or 30x30 grid | agent's location in 22x22 space; range sensor reading about nearby objects | physical features of robot arm; the location of cereal; |
| State Representation | $(x_t, y_t, \sum_{i=1}^{t} r_i)$ | $(\,\text{room}_t, x_t, y_t, \sum_{i=1}^{t} \max(r_i, 0))$ | $(x_t, y_t, z_t$ $\text{roll}_t, \text{pitch}_t, \text{yaw}_t$ $\sum_{i=1}^{t} r_i)$ | same as observation | $(x_t, y_t, \sum_{i=1}^{t} r_i)$ | $(x_t^{\text{rbt}}, y_t^{\text{rbt}}, z_t^{\text{rbt}},$ $x_t^{\text{obj}}, y_t^{\text{obj}}, z_t^{\text{obj}},$ $s_t^{\text{grip}}, \sum_{i=1}^{t} r_i)$ |
| Action | 5 discrete actions: up, down, left, right, noop | 18 discrete actions: noop, fire, left, $\cdots$ | 4 discrete actions: forward, backward, left, right | 2 discrete actions: left, right | $(dx, dy)$ in continuous action space | 7 discrete actions up down, left, right, forward, downward open/close gripper |
| Reward | apple +1 gold +10 rock -0.05 | mostly zero, sparse positive rewards when collecting objects | target +10 | 0 if going left $-\frac{0.01}{10}$ or $-\frac{0.01}{30}$ if going right 1 at the last step if always going right | key +1 door +2 treasure +6 | lift the object +0.5 release the object -0.5 Object in the correct bin +1 |
| Time limit | 45 steps | 4500 steps | 250 steps | 10 or 30 steps | 1000 steps | 1000 steps |
| Stochasticity | random initial location of agent or random initial location of gold or sticky action | random initial delay or sticky action | random initial location in the upper part | deterministic | random normal noise in the agent's initial position | random initial location of cereal |

Table 2: The setting of various environments for our experiments.