[Reviews · NeurIPS 2020]

Review 1

Summary and Contributions: The authors develop an algorithm to counter the problem that often occurs when basing policies off of past trajectories that performed well - in particular, they address the problem of following myopic policies that gave some reward in the past, but not exploring enough to find the optimal policy. They modify the epsilon-greedy algorithm to explore more effectively in order to help alleviate this problem.

Strengths: - I think the work is addressing an interesting and significant problem encountered when basing policies off of past trajectories that performed well. - They appear to have good empirical results for their algorithm.

Weaknesses: - The algorithm description and training procedure was not clear from the main body of the paper. - The problem statement could be more clear (see my notes later regarding section 2.1). - The presentation of the empirical results could be improved.

Correctness: The methodology seems to be correct, but I had difficulty understanding their training algorithm.

Clarity: I thought that the description of the algorithm itself was difficult to understand (not because of poor grammar, but because was not detailed enough.)

Relation to Prior Work: There is a summary of related literature, but I felt like the connection between their work and previous literature was not made explicit enough. For example, I think it would be good to say the goal of your method is not to improve imitation learning in itself, but to employ imitation learning methods to improve exploration.

Reproducibility: No

Additional Feedback: - I think you could improve the discussion of what exactly the role of embeddings e_t and how your training objective is similar to / is different from the typical training objective to maximize total expected reward (section 2.1). You define e_t to be to be a learned state representation (using current and past states). Then you must find the sequence of embeddings that maximize the total sum of rewards (the policy is a function of the sequence of embeddings). However, it's not clear to me that the optimal parameters g^* and \theta^* are identifiable, i.e., there may be many combinations of g and \theta that result in maximizing the total sum of rewards / not necessarily a unique solution. My interpretation is that the embeddings are learning a useful state representation. I think your technique is not optimizing a new objective per say, but using a different optimization method to try to maximize total expected reward objective. - I would also like it to be more clear in the beginning (section 2.1) whether the policy \pi is trained to directly maximize reward or only to best imitate the demonstration. It seems to me that \pi is trained to best imitate the demonstration and g is optimized to find the best demonstration. If this is true, perhaps this could be more explicit in the argmax statement in line 68. - In line 74, it might be nice to briefly mention how you cluster the embeddings. Are these clusters updated over time? - I generally found the section from lines 118-132 hard to understand. I would like to see you write out an algorithm procedure for your method, so that it is more clear what your exact training procedure is. I see that you have more details of the algorithm in the appendix - however, it is still useful to make sure that the reader can understand the main principles of the algorithm from the main body of the text. - I am confused by the definition of u in line 120. Is u indexing an embedding cluster? What does u = -1 mean? - I did not see you define r^{im}. - What does \Delta t refer to in line 122? - In the experiments section, I liked how you set up the section with the goal of answering three questions. I felt that you addressed the answers to these questions but the answers could be made more explicit and clear. For example, for answering question (2) it is addressed in lines 207-214 somewhat, but the figure you address 3d is very small and difficult to interpret.


Review 2

Summary and Contributions: The paper presents a novel goal conditional RL method where the goal is specified as a series of embeddings of the desired target trajectory. The trajectory is sampled from a replay of trajectories which maintained to ensure recency and diversity in the embedding space. This allows the agent to develop a virtual tree of the environment and explore beyond the preipheries this tree. More specifically, the policy attends over the series of embeddings and gets rewards for tracking the trajectory. There is an additional supervised loss which increases the probability of hindsight actions. Experiments are performed in several synthetic maze like environments, on Atari and in Robotics, Deep Sea and MuJoCo environments. In each, the proposed method outperforms the baselines considered which use other forms of exploration bonuses such as pseudo-counts. For Atari however, the method still uses additional count based bonuses.

Strengths: The idea of specifying goals as entire trajectories and not just the final state is novel to the best of my knowledge and is a very interesting avenue. The agent need not remember the exact means to navigate the entire MDP, especially unpromising areas which are very common in exploration, and instead rely on its "parametric memory". This allows the use of policy's capacity more efficiently in learning primitives that are more invariant across the MDP or towards high rewarding areas. The experiments are broad with multiple domains and provide good intuition about the strengths of the method. The authors also do a good job characterizing the method relative to the literature.

Weaknesses: The method still uses an additional supervised loss which seems not clearly motivated and I would have preferred more experiments ablating the need for this. The method uses a parametric memory but compares to on-policy methods mostly. As it also uses an additional imitation loss, comparison with an off-policy method would be more fair. There is quite a bit of engineering that goes into organizing the buffer, with several hyper-parameters involved. Although the authors provide ablations for many of their choices, the number of choices seems to indicate a level of brittleness in the algorithm. The need to additional pseudo-counts for Atari indicates weaknesses in the proposed exploration method which is not fully fleshed out.

Correctness: The paper is correct and the methodology is empirically rigorous.

Clarity: The paper is clearly written. The details of the maintaining the replay buffer could be better organized and presented. The graphs could use more legible labelling.

Relation to Prior Work: The authors do a good job relating this method to previous methods in literature.

Reproducibility: Yes

Additional Feedback: The paper would be stronger with more experiments ablating the value of supervised loss and the pseudo-counts used in atari experiments. The number of hyper-parameters associated with the parametric memory seem quite high and simplifying this would also strengthen the value of the method significantly. The authors could add additional related work from off-policy or model-based exploration literature that also share many of the related features of the proposed method. ===================== Post Author Feedback: Not fully satisfied with the authors response to these questions but I still think this is a good paper and will keep my score unchanged.


Review 3

Summary and Contributions: **Update: I thank the authors for their response, and for including the comparison to the curiosity work that I had requested. The paper proposes an approach for performing long-horizon, sparse reward RL tasks. This involves storing trajectories for reaching various states in a buffer. Exploration is done by sampling states from this buffer and using the corresponding trajectories to guide agent behavior via imitation. Authors show new SOTA performance on Montezuma's revenge and include results on visual navigation and robot bin-picking in simulation.

Strengths: Significance & Novelty : 1) Proposes an effective new approach to exploration in RL, by storing the highest-reward trajectory taken to reach diverse state clusters, which the agent then imitates in order to revisit states (Initally, random sampling of states from the buffer ensures diverse exploration, and the sampling is gradually made more greedy to allow for exploitation). This method leverages imitation signal which enables directed exploration, without requiring expert demonstrations. Importantly, the directed exploration includes states which yield smaller cumulative reward in the short term. 2) Uses a novel policy architecture for imitation which is conditioned on trajectories, and uses attention to weigh the different states in the source trajectory differently. Authors include visualizations that show the agent pays more attention to states ahead of the current state in the source trajectory, highlighting the importance of conditioning on the entire trajectory instead of using vanilla behavior cloning. Emperical Evaluation 1) The paper includes extensive evaluation, on a broad set of environments with sparse rewards where exploration is a challenge, including grid-worlds, atari games, and simulated robotic manipulation and navigation. Comparisons include an ablation without the imitation policy, prior work where only high return trajectories are used for self-imitation, and PPO with count based exploration bonus. Authors also include some results that show ability to generalize in changing MDPs by exploring effectively in grid-world settings, and the ability to handle random initial states for the grid world and robotics tasks.

Weaknesses: Emperical Evaluation: This work does not cite or compare against an important prior work for exploration in deep RL involving intrinsic curiosity [1-2]. While it appears intrinsic curiosity has worse performance on Montezuma's revenge, the comparison must still be included for completeness and intrinsic curiosity must be discussed in the related work. I trust the authors to address this in the rebuttal period. [1] : Curiosity-driven exploration by self-supervised prediction (Pathak et al.) [2] : Large Scale Study of Curiosity-Driven Learning (Burda et al.)

Correctness: The emperical methodology seems correct.

Clarity: The paper is well motivated and clear, and provides a detailed explanation of the methodology and systematic discussion of the various experiments conducted.

Relation to Prior Work: Discussion related to intrinsic curiosity is currently missing, please see the weaknesses section for more details.

Reproducibility: Yes

Additional Feedback:


Review 4

Summary and Contributions: This paper proposes a method to learn a trajectory-conditioned policy that learns effective exploration policy by imitating diverse demonstration trajectories from memory. The paper mainly evaluates the proposed approach on exploration environments with sparse reward setting and also performs other analysis on the overall performance of the algorithm.

Strengths: - The overall idea is novel and interesting. The authors exploit the past demonstration trajectories to find more diverse trajectories which is a more informative way to utilize the past data. - The evaluations are extensive. The paper covers a variety set of evaluations on hard exploration Atari games, robotics tasks, navigation environments, and some other domains. This ensures the generality of the proposed methods. - I enjoy reading the last section of the paper where the authors also evaluate the robustness and generalization properties of the algorithm. The authors also carefully explained the performance gaps among various algorithms.

Weaknesses: - Although the proposed approach overall is reasonable. It is not quite clear whether it is enough to rely on the diversity led by imitation error. As indicated by the author in lines 98 and 99, the novel states are generated when the trained policy does not exactly follow the demonstrations. Does it mean the diversity comes from this error? If so and if the assumption is that the trained policy performs well on what it is trained on, the error will be small and the novel states will be less different from the original state in the demonstration trajectories. If so, what the method will do to prevent the bias led by the original demonstration trajectories? Is this the reason why the approach shows pretty different performance under certain random seed? Please clarify this point. - As pointed out in the strength of the paper, it is nice to see the authors willing to demonstrate the robustness and generalization of the algorithm. However, I also feel that too many things are squeezed together in the experiment section. This leads to less extensive and clear explanations. I understand that the authors are bounded by the total number of pages, but from a good presentation point of view, it will be better if the experiment sections can outline more about the important messages and provide thorough descriptions. - Table 1 is in the wrong scale. You can adjust the scale of the entire table so that it fits the page. Same thing with Equation (1). - The layout of Figure 3 also looks odd and hard to read. Please adjust them and the inline space: the caption is too close to the sub-figures. - In the baseline for Montezuma's Revenge, the authors mention the DTRA+EXP performs poorly. However, the results are not shown in Table 1. What is the performance of the DTRA+EXP?

Correctness: Yes.

Clarity: Yes. Please see other writing / presentation suggestion in the aboev section.

Relation to Prior Work: Yes.

Reproducibility: Yes

Additional Feedback: ===================== Post Author Feedback: I have read the authors' responses and most of my concerns are resolved. Therefore, I will keep my score as acceptance.

[Author Response · NeurIPS 2020]

We thank reviewers for positive feedback, mentioning DTSIL as an effective novel method (R2,3,4) for a significant
problem (R1), extensively evaluated (R1,2,3,4), and systematically discussed (R3). We will incorporate the suggestions.

**[R1] Problem statement:** As R1 interpreted, embeddings are high-level state representations which can differentiate
meaningfully distinctive states. Instead of directly maximizing expected return, we proposed a novel way to find
best demonstrations $g^*$ with (near-)optimal return and train the policy $\pi_\theta(\cdot|g)$ to imitate any trajectory $g$ in the buffer,
including $g^*$. A solution of $g^*$ and $\theta^*$ is not necessarily unique. As stated in L69, DTSIL allows for exploiting
multiple trajectories with the best rewards found during training. We approximately solve the joint optimization
problem $g^*, \theta^* = \arg\max_{g,\theta} \mathbb{E}_{\pi_\theta(\cdot|g)}[\sum_{t=0}^{T} \gamma^t r_t]$ via sampling-based search for $g^*$ over the space of $g$ realizable by
the (trajectory-conditioned) policy $\pi_\theta$ and gradient-based local search for $\theta^*$. We will revise and improve Sec. 2.1 to
make this clear. **Meaning of** $u, \Delta t$ **(L120):** For each episode, $u$ denotes the index of state in the given demonstration
that is lastly visited by agent. The initial value $u = -1$ (at the beginning of episode) means no state in the demonstration
has been visited. $r^{\text{im}}$ is imitation reward with a value 0.1. $\Delta t$ is the number of states in the demonstration to be
compared with $e_{t+1}$ to determine reward for each step $t$ (L123). More details were provided in Appendix B.1, especially
Fig. 2 for illustrations. We will add these pointers and more descriptions in main text to clarify our algorithm. **Related**
**Work:** We will make the connection between DTSIL and prior works more clear, especially for imitation learning part.

**[R1,R2] Embedding clusters:** Pseudocode for organizing clusters was in Appendix A.3. We will add this pointer
in L74 and a brief explanation: In the buffer, we keep a representative state embedding for each cluster. If a state
embedding $e_t$ in the current episode is close to a representative state embedding $e^{(k)}$, we increase visitation count $n^{(k)}$
of the $k$-th cluster. If the sub-trajectory $\tau_{\leq t}$ of current episode up to step $t$ is better than $\tau^{(k)}$, $e^{(k)}$ is replaced by $e_t$.

**[R2] Supervised learning:** With SL objective, we leverage the *actions* in demonstrations, similarly to behavior cloning,
to help RL for imitation of diverse trajectories. DTSIL+EXP without SL performs worse on Montezuma's Revenge
(MR) and Pitfall where imitation is difficult due to many obstacles and dangers (Tab. A). **Pseudo-count bonus:** DTSIL
discovers novel states mainly by random exploration after the agent finishes imitating the demonstration. The pseudo-
count bonus brings improvement over random exploration by explicitly encouraging the agent to visit novel states.
Prior works (e.g. CoEX, NGU) commonly use a count-based bonus for exploration (EXP). DTSIL is complementary
to EXP; combining both performs better than DTSIL (Tab. A) and PPO+EXP (Tab. 1). We will add the ablative
study. **Hyper-parameters:** Assume agent's location in state embeddings is normalized to $[0, 1]$ for each coordinate
and the distance metric is $l_\infty$. When clustering embeddings in parametric memory, $\delta = 0.1$ will discretize 2D location
space into $\sim$10×10 grid, an intuitively reasonable size. We can remove a hyper-parameter $\Delta t$ by setting $\Delta t = m$,
because the larger $\Delta t \in [1, m]$ leads to better performance (Appendix E.1). DTSIL(+EXP) with $\Delta t = m = 40, \delta = 0.1$
achieves scores 8.2 (Apple-Gold), 21365 (MR), 10192 (Pitfall), 1915 (Venture), 7.6 (navigation), 56 (manipulation with
discrete actions), comparable with numbers we reported in submission. Thus, DTSIL with a single hyper-parameter
setup can perform robustly well and not brittle across various domains. **Off-policy methods:** We listed off-policy
methods A2C+SIL and NGU in Tab. 1 in the submission. We additionally run R2D2[1] on Atari (Tab. A) and HER[2]
on robotics manipulation with high-dimensional continuous actions, where DTSIL gets a score 20 but HER gets 0.
Many off-policy methods tend to discard old experiences with low rewards and hence may prematurely converge to
sub-optimal behaviors, but DTSIL using these diverse experiences has a better chance of finding higher rewards in the
long term. We will add this comparison and more discussions about off-policy and model-based exploration methods.

**[R3]** We will cite Pathak et al. & Burda et al. as related works and
add more discussion: Intrinsic curiosity uses the prediction error as
intrinsic reward signals to incentivize visiting novel states, whereas
DTSIL instead imitates long trajectories in diverse directions, which
can lead to deeper exploration. As R3 suggested, we show additional
experiments of ICM[3] and RF[4] for 800M steps (Tab. A).

| | DTSIL+EXP | DTSIL+EXP w/o SL | DTSIL | ICM | RF | R2D2 |
|---|---|---|---|---|---|---|
| MR | 26,314 | 10,112 | 5,712 | 100 | 10,200 | 400 |
| Pitfall | 11,875 | 1,966 | 2,436 | 0 | 0 | 0 |
| Venture | 2,135 | 1,898 | 1,482 | 1,813 | 1,859 | 1,997 |

**Table A:** Comparison with variants of DTSIL and additional baselines, one run for each baseline due to limitation of computational resources. We will report results of more runs in the revision.

**[R4] Diversity:** DTSIL's ability to find diverse states does not rely solely on the "imitation error". After visiting the
last (non-terminal) state in the demonstration, the agent performs random exploration (because $r_t^{\text{DTSIL}} = 0$) around
and beyond the last state until the episode terminates, to push the frontier of exploration. We prevent "the bias led by
original demonstrations" by allowing flexibility in following them and replacing them with better trajectories. Different
performances under different random seeds are due to huge positive rewards in some states on MR and Pitfall. Once the
agent luckily finds these states in some runs, DTSIL can exploit them and perform much better than other runs.

**[R1,R4] Presentation:** The important messages about the experiments were summarized as three questions at the start
of Sec. 4. Per R1's comments, we will explicitly connect questions and experimental results in the revision. We will
also emphasize these take-away messages and point to thorough descriptions in Appendix C, as R4 suggested.

**[R1,R2,R4]** Fig. 3d shows that trajectory-conditioned policy imitates diverse demonstrations well with proper attention
weights. Fig. 4 shows DTRA+EXP. We will adjust Tab. 1 & Fig. 3 as suggested and use more legible labels in graphs.

## Footnotes

[1] Kapturowski et al. Recurrent experience replay in distributed reinforcement learning. [2] Andrychowicz et al. Hindsight experience replay. [3] Pathak et al. Curiosity-driven exploration by self-supervised prediction. [4] Burda et al. Large-scale study of curiosity-driven learning.


[Meta-Review · NeurIPS 2020]

The paper presents an approach to deal with sparse reward setups by storing high-reward trajectories/states from the past experience and use them to perform a more directed exploration. The idea of reusing past trajectories to direct exploration at future timesteps has been attempted several times in the literature but this paper finally seems to get it right without relying on the environment being deterministic or need for expert demonstration trajectories. All the reviewers liked the idea of the paper and but had concerns regarding baseline comparisons. The authors' provided the rebuttal and addressed some of the concerns. Given the initial reviews and author's rebuttal, reviewers agreed that the paper provides sufficient insights to be accepted. However, the baseline comparisons still need to be improved for the final version, for instance, comparison to off-policy and model-based RL methods, etc. Please refer to reviewers' final comments and address their concerns in the camera-ready.